

# Study of Arabian Red Sea coastal soils as potential mineral dust sources

P. Jish Prakash[1], Georgiy Stenchikov[1], Weichun Tao[1], Tahir Yapici[1], Bashir Warsama[1], and Johann Engelbrecht[2,1]

[1]King Abdullah University of Science and Technology (KAUST), Physical Science and Engineering Division (PSE), Thuwal, 23955-6900, Saudi Arabia.

[2]Desert Research Institute (DRI), Reno, Nevada 89512-1095, U.S.A.

*Correspondence to*: P. Jish Prakash (jishprakash@gmail.com)

**Abstract**. Both Moderate Resolution Imaging Spectroradiometer (MODIS) and Spinning Enhanced Visible and InfraRed Imager (SEVIRI) satellite observations suggest that the narrow heterogeneous Red Sea coastal region is a frequent source of airborne dust that, because of its proximity, directly affects the Red Sea and coastal urban centers. The potential of soils to be suspended as airborne mineral dust depends largely on soil texture, moisture content, and particle size distributions. Airborne dust inevitably carries the mineralogical and chemical signature of a parent soil. The existing soil databases are too coarse to resolve the small but important coastal region. The purpose of this study is to better characterize the mineralogical, chemical and physical properties of soils from the Red Sea Arabian coastal plane, which in turn will help to improve assessment of dust effect on the Red Sea and land environmental systems and urban centers. Thirteen surface soils from the hot-spot areas of wind-blown mineral dust along the Red Sea coastal plain were sampled for analysis. Analytical methods included Optical Microscopy, X-ray diffraction (XRD), Inductively Coupled Plasma Optical Emission Spectrometry (ICP-OES), Ion Chromatography (IC), Scanning Electron Microscopy (SEM), and Laser Particle Size Analysis (LPSA). We found that the Red Sea coastal soils contain major components of quartz and feldspar, as well as lesser but variable amounts of amphibole, pyroxene, carbonate, clays, and micas, with traces of gypsum, halite, chlorite, epidote and oxides. The wide range of minerals in the soil samples was ascribed to the variety of igneous and metamorphic provenance rocks of the Arabian Shield forming the escarpment to the east of the Red Sea coastal plain. The analysis revealed that the samples contain compounds of nitrogen, phosphorus and iron that are essential nutrients to marine life. The analytical results from this study will provide a valuable input into dust emission models used in climate, marine ecology, and air-quality studies.

## 1 Introduction

Mineral dust is the most abundant atmospheric aerosol, primarily suspended from ground in arid and semi-arid regions of the globe (Buseck et al., 1999;Washington and Todd, 2005;Goudie, 2006;Muhs et al., 2014), including deserts of the Arabian Peninsula (Edgell, 2006). Dust aerosols profoundly affect climate, biogeochemical cycles in the ocean and over land, air-quality, atmospheric chemistry, cloud formation, visibility, and human activities (Prospero et al., 2002;Haywood and Boucher, 2000;Hsu et al., 2004;Sokolik and Toon, 1999;Kumar et al., 2014;De Longueville et al.,





2010;Jickells et al., 2005;Mahowald, 2009;Huang et al., 2006;Huang et al., 2014;Fryrear, 1981;Nihlen and Lund, 1995;Hagen and Woodruff, 1973;Bennett et al., 2006;Bennion et al., 2007;Twomey et al., 2011;Wang et al., 2010). The Arabian Peninsula is one of Earth's major sources of atmospheric dust, which contributes as much as 11.8% (22 – 500 Mt/a) of the total (1,877 – 4000 Mt/a) global dust emissions (Tanaka and Chiba, 2006). The Red Sea surrounded

by African and Arabian deserts is strongly affected by dust. Along with profound impact on the surface energy budget over land and over the Sea (Brindley et al., 2015;Kalenderski et al., 2013;Osipov et al., 2015), dust is an important source of nutrients especially for the oligotrophic northern Red Sea region (Acosta et al., 2013). From preliminary observations it is estimated that 5 to 6 major dust storms per year impact the region, depositing about 6 Mt of mineral dust into the Red Sea (Prakash et al., 2015). Simulations and satellite observations suggest that the coastal dust

contribution to the total deposition flux into the Red Sea could be significant even during fair weather conditions (Jiang et al., 2009). However, the mineralogy and chemical composition of dust generated from the Red Sea coastal region remain uncertain. The coastal plain is a narrow highly heterogeneous piedmont area and existing soil databases do not have enough spatial resolution to represent it adequately (Nickovic et al., 2012).

To explain the connection between soil properties and airborne dust abundance and composition we below discuss the

physically-based dust generation parameterizations currently used in the advanced modeling systems (Grell et al., 2005;Zender et al., 2003).  The vertical mass flux of dust size component j, $F_j$ (kg m$^{-2}$ s$^{-1}$), generated from the ground into the atmosphere could be calculated in the following way:

$$F_j = TS f_m \alpha Q_s \sum_{i=1}^{I} M_{ij} \qquad\qquad (1)$$

Where, $T$ is a tuning constant for adjusting to different horizontal and temporal resolutions. The parameter $f_m$ is a grid

cell fraction of exposed bare soil suitable for dust mobilization. The coefficient "$\alpha$" is sandblasting mass efficiency determined by the mass fraction of clay particles in the soil.  "$M_{i,j}$" is the mass fraction of each source mode $i$, carried in each transport bin $j$. The parameter "$S$" is the erodibility factor that accounts for the susceptibility of a landscape to wind erosion controlled by the non-erodible roughness elements and the erodibility of soils within the erodible area of a landscape (Webb and Strong, 2011). In dust emission models, the soil erodibility control is represented through

the effects of soil texture and moisture content on the threshold friction velocity $u^*_t$ and the aerodynamic roughness length $f_z$ (Oleson et al., 2010;Webb and Strong, 2011). The parameter $S$ is often defined via the so-called "source function" that accounts for the spatial distribution of dust source intensities based on a variety of algorithms (Menut et al., 2013). The parameter $Q_s$ is the total horizontally saltation mass flux (kg m$^{-1}$ s$^{-1}$), which is proportional to the third power of friction velocity ($u^*_s$) when it exceeds a threshold velocity $u^*_t$ (Oleson et al., 2010;Zender et al., 2003).

Equation (1) relates the size-dependent soil dust properties with that emitted in the atmosphere, where dust size-distribution and compositional characteristics are further adjusted as dust particles atmospheric residence time depends on their sizes and weights. The atmospheric dust size distribution and mineralogical/chemical composition defines radiative, ecological, and health effects of dust.  The importance of keeping track of dust mineralogy during the atmospheric transport was recently recognized and implemented in the models (Perlwitz et al., 2015b, a).





Minerals previously found in continental soils from dust generation regions include quartz, feldspars, calcite, dolomite, micas, chlorite, kaolinite, illite, smectite, palygorskite, mixed-layer clays, iron oxides, gypsum, and halite (Pye, 1987;Scheuvens and Kandler, 2014;Goudie, 2006;Engelbrecht and Moosmüller, 2014). Al-Farraj (2008) studied the soils from the Jazan region of southern Saudi Arabia, identifying smectite, kaolinite and illite as the predominant clay

minerals, together with lesser amounts of chlorite, quartz and feldspars. Shadfan et al. (1984) investigated mineralogical content and general characteristics of soils from some agricultural areas in Saudi Arabia. They found carbonate, quartz and gypsum to be the main constituents of the sand and silt fractions in soils of the eastern region, while quartz, carbonate and feldspars dominate soils in the central region. The soils in the west contain mainly quartz, feldspars, hornblende and mica. Palygorskite was found to be the main clay mineral in soils in the eastern region,

kaolinite in the central region, and kaolinite, smectite and mica in the western region. Aba-Husayn et al. (1980) mineralogically analyzed  soils from  the southwestern region of Saudi Arabia, along the mountainous Asir region between Mecca and Abha. They found major amounts of quartz, feldspars and micaceous minerals in the silt fractions, with the clay-size fractions of kaolinite, smectite, and vermiculite, with kaolinite in the well-drained highland areas. Viani et al. (1983) studied fourteen soils from alluvial basins in the Wadi ad Dawasir, and Wadi Najran areas of

southwestern Saudi Arabia. Due to the fact that the alluvial clay-size fractions were from weathered igneous rocks of the surrounding mountains, they were found to be composed largely of smectite, mica, kaolinite, chlorite, palygorskite and vermiculite. A similar study on soils of the eastern region of Saudi Arabia (Lee et al., 1983) found smectite, palygorskite, kaolinite, chlorite, mica and vermiculite in the clay-size fractions. The fallen dust along the tracks of dust storms within major deserts in the world were collected and analyzed by Al-Dousari and Al-Awadhi (2012). They

showed that fallen dust from eastern zones (Taklimakan, Gobi, and Australian deserts) are characterized by higher percentage of feldspar and clay minerals in comparison to the western zones (Sahara and Arabian deserts) and western Sahara desert dust is differentiated by the highest average quartz percentage (66%). Al-Dousari and Al-Awadhi (2012) showed dust palls and sand dunes in Iraq to be composed of quartz, feldspar, calcite, gypsum, dolomite and heavy minerals. Al-Dabbas et al. (2012) analyzed dust samples over Iraq and showed the minerals as quartz (58.6%),

feldspars (17.3%), calcite (15.4%), and small amount of gypsum (5.5%). They also recognized clay minerals (chlorite, illite, montmorillonite, palygorskite and kaolinite).

The sample area in this study lies within the approximately 60 – 70 km wide Tihāmah coastal plain, comprised of the Tihāmat Asīr in the south and the Tihāmat Al-Hejaz to the north. The plain is bounded by the Red Sea in the west, with the mountains of Midyan, Ash Shifa and Asir forming an escarpment to the east (Edgell, 2006), with few breaks

in the mountains in the northwest. The mountains form a 1,000 – 3,000 m elevation Red Sea escarpment, comprised of igneous, metamorphic and volcanic rocks of variable age, from Pre-cambrian (1,000 – 545 million years) to the less than 30 million years in age (Grainger, 2007). The Red Sea rift basin itself is overlain by the much younger sediments of Quaternary age (< 2.6 million years).

With the exception of the area around Jazan in the south, which is impacted by the Indian Ocean monsoon, the Red

Sea coastal region has a desert climate characterized by extreme heat, reaching 39 °C during the summer days, with a drop in night-time temperatures of about 10 °C.  Although the extreme temperatures are moderated by the proximity





of the Red Sea, in summer the humidity is often 85% or higher during periods of the northwesterly *Shamal* winds. Rainfall diminishes from an annual average of 133 mm at Jazan to 56 mm at Jeddah, and 24 mm at Tabuk in the north. Vegetation is sparse, being restricted to semi-desert shrubs, and acacia trees along the ephemeral rivers (wadis), providing forage for small herds of goats, sheep and dromedary camels.

5 During infrequent but severe rainstorms, run-off from the escarpment along wadis often produce flash floods. With such events, fine silt and clays are deposited on the coastal plain, which are transformed into dust sources during dry and windy periods of the year. The resultant dust is transported and deposited on the coastal plain and adjacent Red Sea by prevailing northwesterly to southwesterly winds, with moderate breezes (wind speed >5.5 m/s) from the north (http://www.windfinder.com/weather-maps/report/saudiarabia#6/22.999/34.980).

## 10  2 Objectives

This study aims to provide mineralogical and chemical compositions of thirteen surface soils collected at four areas within the central part of the Red Sea coastal plain of Saudi Arabia, (Fig. 1). The dust hot spots are located within the narrow coastal region, and because of their proximity to the Red Sea, contribute to the dust/nutrient balance of the Sea, during both dusty and fair weather conditions. The coastal plains of the Arabian Peninsula along the Red Sea and

15 Persian Gulf are among the most populated areas in this region hosting the major industrial and urban centers. Airborne dust profoundly affects human activities, marine and land ecosystems, climate, air-quality, and human health. The observations suggest that the narrow Red Sea coastal belt is an important dust source region, augmented by the fine-scale sediment accumulations, scattered vegetation, and varying terrain. Limited compositional information is available on soils along the Red Sea coastal region.

The present study examines soil mineralogical and chemical compositions, and individual particle morphology from the samples taken at the hot-spot areas. This will help to better quantify the ecological impacts, health effects, damage to property, and optical effects of dust blown from these areas (Engelbrecht et al., 2009a, b;Weese and Abraham, 2009). The mineralogical compositions of the soils tie into that of the parental rocks, weathering conditions and time. This research will complement soil and dust studies performed in the Arabian Peninsula as well as globally

(Engelbrecht and Moosmüller, 2014;Engelbrecht et al., 2009b). Knowledge of the mineralogy of the soils will provide data on refractive indices, particle size and shape parameters, which can be used to calibrate dust transport models, and help to assess the impact of dust events on the coastal plain and the Red Sea.

## 3 Sampling and analysis

A total of thirteen samples were collected at four localities along the Red Sea coastal plain (Fig. 1). Three samples

(S1–S3) collected at 25 km northeast of Mastorah near washland of Wadi Hazahiz located 26 km from Red Sea. Samples (S4–S6) collected at 30 km east of Ar Rayis near Ushash, which is a village in Al Madinah province located 32 km from Red Sea. Samples (S7–S9) collected at 27 km north of Yanbu at washland of Wadi al Wazrah with an





elevation of 158 m above sea level and located 30 km from Red Sea. Four samples (S10–S13) collected at 28 km southwest of Mecca near Wadi An Numan located 45 km from Red Sea. The coordinates of the sample sites are provided in Table 1. All thirteen samples can be classed as Leptosols (Regosols) (http://www.fao.org/ag/agl/agll/wrb/soilres.stm).

The assumption is that at least part of the dust in the ambient atmosphere in this coastal region is from windblown and otherwise disturbed soils along the Red Sea coast. Jiang et al. (2009) and Kalenderski et al. (2013) found that the coastal area emits about 5–6 Mt of dust annually. Due to its close proximity, a significant portion of this dust is likely to be deposited to the Red Sea, which could be comparable in amount to the estimated annual deposition rate from remote sources during major dust storms (Prakash et al., 2015).

The grab soil samples collected in the field were sieved to D<1 mm to remove pebbles, plant material and other unwanted artifacts. Where necessary, they were air-dried in the laboratory, before being labeled, catalogued and stored in capped plastic bottles. Sub-sets of these samples were screened to D<38 µm for mineral analysis by X-ray powder diffraction (XRD), chemical analysis, and Scanning Electron Microscopic (SEM) based individual particle analysis. Further samples of 75 µm < D < 125 µm were sieved for mineralogical investigation by optical microscopy, and D <
600 µm for Laser particle size analysis (LPSA).

Petrographic microscopy is particularly suited to the optical identification of mineral grains larger than about 10 µm (Kerr, 1959). It remains a cost effective and accurate technique to obtain mineralogical information which is otherwise difficult to obtain, e.g. the identification of feldspars, amphiboles and pyroxenes. The 75 µm < D < 125 µm sieved soil fraction grains were mounted in epoxy on a glass slide and ground to a thickness of approximately 30 µm, for
transmitted light optical microscopy. Mineral optical properties such as texture, color, pleochroism, birefringence, relief, and twinning were used to identify silicate minerals and to estimate their abundance in the samples. Optical properties such as scattering, absorption and refractive indices vary, depending on the mineralogical content of the dust in the atmosphere.

X-ray diffraction (XRD) is a non-destructive technique for characterization of minerals, including quartz, feldspars,
calcite, dolomite, clay minerals, and iron oxides, particularly for the fine soil and dust fractions. Dust reactivity in the seawater as well as optical properties depend on their mineralogy, e.g. carbonates and some silicates are generally more soluble in water than for example feldspars, amphiboles, pyroxenes or quartz. A Bruker D8® X-ray powder diffraction (XRD) system was used to analyze the mineral content of the soil samples. The diffractometer was operated at 40 kV and 40 mA, with Cu Kα radiation, scanning over a range of 4° to 50° 2θ.  The Bruker Topas® software and
relative intensity ratios were applied for semi-quantitative XRD analyses of the D < 38 µm screened dust samples (Rietveld, 1969;Chung, 1974;Esteve et al., 1997;Caquineau et al., 1997;Sturges et al., 1989).

Laser particle size analysis (LPSA) was performed on the thirteen soil samples. The LPSA system measures the size-class fractions of a soil or sediment sample in an aqueous suspension, based on the principle that light scatters at angles inversely proportional to, and with intensity directly proportional to particle size (Gee and Or, 2002). Dust emission



rates from soils and sites of airborne particles strongly depend on the soil particle size distributions. The optical properties of airborne particles, such as scattering and absorption, depend on their particle sizes. The grab samples were sieved to D < 600 µm before being introduced to the laser analyzer (Micromeritics Saturn DigiSizer 5200®) in an aqueous medium of 0.005% surfactant (sodium metaphosphate). The suspensions were internally dispersed by

applying ultra-sonication and circulated through the path of the laser light beam. The measured size-class fractions were grouped as clay (D < 2 µm), silt (2 µm < D < 62.5 µm) and sand (62.5 µm < D < 600 µm), (Engelbrecht et al., 2012).

The D < 38 µm sieved samples were chemically analyzed for elemental composition by Inductively Coupled Plasma Optical Emission Spectrometry (ICP-OES), and their water soluble ions by Ion Chromatography (IC). For ICP-OES,

splits of 0.1g of each of the samples were digested in a 1:3:1 mixture of concentrated hydrofluoric acid (HF), hydrochloric acid (HCl) and nitric acid ($HNO_3$), in a microwave oven (Milestone Ethos1®) operated at a temperature up to 195 °C for 15 minutes. The solutions were diluted from 25 ml to 250 ml before being analyzed on a ICP-OES (Varian 720-ES®), for sodium (Na), magnesium (Mg), aluminum (Al), silicon (Si), phosphorus (P), sulfur (S), potassium (K), calcium (Ca), titanium (Ti), vanadium (V), chromium (Cr), manganese (Mn), iron (Fe), cobalt (Co),

nickel (Ni), copper (Cu), zinc (Zn), strontium (Sr), cadmium (Cd), barium (Ba) and lead (Pb). The accuracy of the analyses was monitored by analyzing the National Institute of Standards and Technology (NIST) standard reference material 1646a with each batch of soil samples.

Further splits (approx. 0.01 g) of the D < 38 µm sieved samples were sonicated in 15 ml of de-ionized distilled water, the suspension left to settle overnight, and the extractions analyzed by IC (DIONEX ICS-3000®). The water soluble

cations of sodium ($Na^+$), potassium ($K^+$), calcium ($Ca^{2+}$) and magnesium ($Mg^{2+}$), and anions of sulfate ($SO_4^{2-}$), chloride ($Cl^-$), phosphate ($PO_4^{3-}$) and nitrate ($NO_3^-$) were analyzed by this method.

Electron microscopy provides information on the individual particle size, shape, chemical composition, and mineralogy of micron-size particles, important for determining the optical parameters for modeling of dust (Moosmüller et al., 2012). The individual particle chemistry, especially of the soluble minerals such as carbonates, is

often of importance in medical geology and to marine life. The Scanning Electron Microscope (SEM) based individual particle analysis was performed on the D < 38 µm sieved sample splits. A dual approach was followed, the first being the computer controlled scanning electron microscopy (CCSEM) and the second, secondary electron imaging by high resolution scanning electron microscopy (SEM). For each sample, a portion of soil was suspended in isopropanol and dispersed by sonication. The suspension was vacuum filtered onto a 0.2 µm pore size polycarbonate substrate. A

section of the substrate was mounted onto a metal SEM stub with colloidal graphite adhesive. The sample mounts were sputter-coated with carbon to dissipate the negative charge induced on the sample by the electron beam. The automated CCSEM analysis was conducted on a Tescan MIRA 3® field emission scanning electron microscope (FE-SEM). The CCSEM analysis was performed by rastering the electron beam over the sample while monitoring the resultant combined backscattered electron (BE) and secondary electron (SE) signals. The BE intensities were applied

to set grayscale levels, to distinguish particles of interest from background. The system was configured to



automatically measure the size and the elemental composition for about 2,000 individual particles of $0.5 > D < 38\,\mu m$ sizes. Individual particles were classified into particle types according to their elemental compositions. A digital image was acquired of each particle for measurement, and stored for subsequent review. Size measurements were based on diameters obtained from the projected area of each particle, by tracing their outer edges. Compositional information

was determined through collection and processing of characteristic X-rays by energy dispersive spectroscopy (EDS) using a silicon drift detector (SDD). The elements identified in the spectrum were processed to obtain their relative concentrations. The particles were grouped into "bins" by their particle size and chemical ratios. From the chemical measurements, and a priori knowledge of the sample mineralogy (from optical microscopy and XRD), the mineralogy of individual particles can often be inferred, e.g. Si particles being quartz, Ca particles being calcite, Ca plus S particles

being gypsum.

The field emission electron source allows for high magnifications and sharp secondary electron images (SEI). This technique allows for the detailed study of particle shape, surface features, and chemical compositions. Approximately five SEI's with energy dispersive spectra (EDS) for each of the thirteen samples were collected. Fig. B3 (Appendix B) shows SEM secondary electron images and EDS spectra of $D < 38\,\mu m$ soil particles from the sampling site.

**4 Results**

**4.1 Particle Size Analysis**

Particle volume size plots of the $D < 600\,\mu m$ sieved samples are listed in Table 2 and graphically presented in Fig. 2. All thirteen soils are composed of on average close to 89% sand fractions. Also, the silt makes up approximately 10% and the clay on average less than 1.5% of the sample volume.

Field and laboratory measurements on dust from the western U.S.A. (Engelbrecht et al., 2012) showed that dust emissions are largely controlled by their soil particle size distributions (Kok, 2011b, a). It was established that surface soils with a silt content of greater than about 50% and a clay content of less than about 10%, i.e. samples in the "silt loam" field (Fig. 2) have the greatest potential to become re-suspended in the air and to generate airborne mineral dust. This particle size criterion provides an important measure for whether a site or region has the potential to be a

significant dust source (Greely and Iversen, 1985). These include soils from previously identified dust sources such as Bodélé Depression (Washington et al., 2003), loess along the Danube River valley, Kuwait, China (Engelbrecht and Moosmüller, 2014), silt deposits collected on Fuerteventura Island assumed to contain dust from the western Sahara (Menéndez et al., 2014), as well as one diatomaceous silt sample from Reno, USA. Besides the particle size distribution it was shown that moisture content and surface roughness play important roles in the saltation and de-

segregation of soil particles (Marticorena, 2014). Judging from their particle size distributions alone, soil samples collected from the coastal zone of Saudi Arabia are not considered to contain enough silt-size particles to be efficient emitters of dust. However, the satellite images show that these coastal dust sources are activated quite frequently.





### 4.2 Optical microscopy

Mineralogical investigation by optical microscopy of three 75 µm < D < 125 µm sieved samples showed them to consist of partially weathered angular mineral grains in sediments probably aeroded from the Pre-Cambrian basement and Tertiary volcanic rocks of the Arabian Shield, approximately 50 km to the east of the Red Sea coastline (Edgell,

2006). The major minerals identified in this size range are feldspar (mainly plagioclase), quartz, pyroxene (aegerine-augite), amphibole, and mica (biotite, muscovite). Lesser amounts of potassium feldspar (orthoclase, microcline), carbonates (calcite, dolomite), chlorite, epidote and oxides were identified by optical microscopy.

### 4.3 XRD mineral analysis

XRD analysis of the thirteen, D<38µm sieved samples from the Red Sea coastal plain (Fig. 3) confirmed variable

amounts of quartz (19 – 44%) and feldspars (plagioclase, K-feldspar) (31 – 48%), with variable amounts of amphibole (and pyroxene) (4 – 31%), lesser amounts of calcite (0.4 – 6.2%), dolomite (1.9 – 6.6%), clays and chlorite (smectite, illite, palygorskite, kaolinite) (3.3 – 8.3%), with traces of gypsum (0 – 0.6%) and halite (0.2 – 4.8%). For this and other localities, the mineralogy resembles that of the igneous and metamorphic rocks of the adjacent mountainous escarpment and Arabian Shield (Edgell, 2006). The average amphibole (plus pyroxene) content for the four samples

taken at the southernmost locality (Fig. 1, S10 – S13) is substantially higher than for the nine samples taken at the other three localities (Fig. 1, S1 – S9), being approximately 26% for the former and 11% for the later. This can be attributed to differences in the mineral composition of the Arabian Shield rocks, distance of the sampling sites from the source regions, and the extent of weathering in the surface soils.

### 4.4 Chemistry (ICP-OES and IC)

Chemical analysis of the D < 38 µm sieved bulk samples by ICP-OES and IC are presented in Tables C1&C2 (appendix C) and a plot of the major elements expressed as oxides, shown in Fig 4.  The soils are of consistent chemical compositions throughout the sampled region.

The sedimentary samples all contain major percentages of $SiO_2$, varying between 63% and 78% in the thirteen samples, mostly as the mineral quartz, and lesser amounts of $Al_2O_3$, $CaO$, $Na_2O$, and $K_2O$, in plagioclase and potassium

feldspars. $SiO_2$ together with $Al_2O_3$, $Fe_2O_3$, $TiO_2$, $MnO$, $MgO$, and some $K_2O$ is also contained in the previously identified amphiboles, clays and micas. Small amounts of $CaO$ (0.9 – 1.7%) are contained in gypsum and calcite, and together with $MgO$ (2.3 – 3.1%) in dolomite.

The water soluble ions account for a small percentage of the total mass of the soils, varying between 0.1% and 0.7% for the total cations, and 0.03% and 0.8% for the total anions. These account primarily for calcite and dolomite (~

0.3%), and gypsum (~ 0.2%), with lesser amounts of halite and other chlorides from sea salt. This unexpectedly low concentration of halite and other soluble salts in the soils of the coastal plains can be ascribed to the fact that all the samples were collected at distances varying between 21 and 42 km from the Red Sea coast, and the absence of local playas or other saline soils close to the four sampling areas. It is also expected that the salts had been leached from the



soil samples collected from surface. Also of importance to dust borne nutrients likely to be deposited in the Red Sea is the low concentration of water soluble $PO_4^{3-}$ (avg. 0.003 %) in comparison to the total $P_2O_5$ (avg. 0.4%) in the soils. The phosphorus is largely bound in the low soluble mineral apatite, commonly found in the sediments throughout the Arabian Peninsula.

**4.5 SEM chemical analysis**

Approximately 2000 individual dust particles in the 0.5 – 38 µm size range were analyzed automatically by CCSEM, for chemical composition, particle morphology and size. The particles were classed into 14 bins as per their chemical compositions. Mineral labels were assigned to these chemical bins, e.g. Fe-rich as hematite ($Fe_2O_3$) (also possibly goethite, magnetite or ferrihydrite), Ca-S rich as gypsum ($CaSO_4.2H_2O$), Ca-Mg rich as dolomite ($CaMg(CO_3)_2$), Ca rich as calcite ($CaCO_3$), Ca-Al-Si rich as anorthite ($CaAl_2Si_2O_8$), Na-Al-Si rich as albite ($NaAlSi_3O_8$), K-Al-Si rich as K-feldspar ($KAlSi_3O_8$), and Si-rich as quartz ($SiO_2$). The CCSEM results for the 0.5 – 38 µm analyzed set as well as the 0.5 – 2.5 µm (fine) subset are presented in Fig. 5 and 6.

For the total data set, the samples in the 0.5 – 38 µm size range contain about 0.1 – 10.2% Si (quartz), 5 – 54% feldspar, 45 – 72% clay minerals, as major components with lesser amounts of calcite, dolomite, gypsum, and iron oxides. The clay minerals can occur as individual minerals but largely as coatings on other silicates (Engelbrecht et al., 2009a). The 0.5 – 38 µm set shows a substantial variability in chemical composition, but no distinct differences between the samples within the four localities. The 0.5 – 2.5 µm (fine) subsets of the three samples (S7, S8, and S9) are different from the others in their higher Fe-rich (goethite, hematite) and carbon (carbonates) components, and corresponding smaller amounts of clay (Fig. 6). This can be ascribed to a local difference in the mineralogical composition of the undifferentiated source rocks (Edgell, 2006), as well as weathering conditions.

The size and shapes of the thirteen, D < 38 µm sieved samples are given in Tables C1&C2 (appendix C), with the size distributions graphically displayed in Fig. B1&B2 (appendix B). Fig. 7 shows the average particle size distributions, as well as size and shape (aspect ratio) statistics for D < 38 µm sieved samples, as measured by scanning electron microscopy (CCSEM). For individual samples, the particle sizes are approximately log normally distributed (skewness = 2.3 – 5.5), often showing a slight bimodality, with a small maximum (approx. 12 µm) on the high end of the distributions. The latter can be ascribed to harder, larger silt size particles of quartz and feldspars. The greatest number of particles are tightly clustered about their mean diameters, resulting in high but variable kurtosis values (4.6 – 44.0). The geometric mean diameters for the particles lie in the small range of 2.1 to 3.7 µm, implying similar mineralogy and hardness, and containing sheet silicates such as clays, chlorites, and micas. The mean aspect ratios of the particles also fall in a tight range of 1.40 to 1.48, with a mean value of 1.43.

**5 Discussion and Conclusions**

Mineral dust generated by wind erosion of soils is the major contributor to global aerosol mass loading and column optical thickness. It is especially abundant in the desert and semi-desert regions. Dust affects marine life providing



nutrients to the marine environment and controlling incoming solar and terrestrial radiation. Soils in arid regions are most susceptible to wind erosion, where particles are only loosely bound to the surface by the low soil moisture. Dust uplifting occurs in a source region when the surface wind speed exceeds a threshold velocity (Gillette and Walker, 1977), which is a function of surface roughness elements, grain size, and soil moisture (Marticorena and Bergametti,

1995;Wang et al., 2000). Fine soil particles that can be transported over large distances are released by saltating coarse sand particles (Caquineau et al., 1997). So soil morphology, mineralogy, and chemical composition define the abundance and composition of airborne dust, however, not directly but through the series of complex fine-scale non-linear processes.

The impact of soil dust from natural and anthropogenic sources on climate and air quality has been recognized on a

global scale (Sokolik and Toon, 1996;Tegen and Fung, 1995). However, the regional fine-scale processes of mineral dust emissions and their effect on the environmental processes and human health are poorly quantified in the study region because the spatial distribution of detailed mineralogical, physical and chemical properties of the surface soils at coastal dust source regions ("hot-spots") were not available.

From satellite images we identified four Red Sea coastal areas from which dust was frequently emitted (Jiang et al.,

2009;Kalenderski et al., 2013). The thirteen soil grab samples were collected from these areas for analysis and their mineralogy, chemical composition and particle size distributions were studied. We found that the Red Sea coastal samples collected in this study contain major components of quartz and feldspar (plagioclase, orthoclase), as well as lesser but variable amounts of amphibole (hornblende), pyroxene (aegerine-augite), carbonate (calcite, dolomite), clays (illite, palygorskite, kaolinite, smectite), and micas (muscovite, biotite), with traces of gypsum, halite, chlorite,

epidote and oxides. The range of identified minerals is ascribed to the variety of igneous and metamorphic provenance rocks along the escarpment to the east of the Red Sea coastal plain (Edgell, 2006). Similarly high fractions of quartz and feldspars were reported for Kuwait (Engelbrecht et al., 2009b) and to a lesser extent for Tallil, Tikrit and Taji in Iraq. The samples from the Red Sea coastal region of Saudi Arabia differ substantially from those from Afghanistan, Qatar, UAE, Iraq and Kuwait in that they contain substantially less calcite. They also contain far less dolomite than

the sample from Al Asad in Iraq. These deviations in composition could be ascribed to differences in provenance and geology. The coastal plain is bounded by the Red Sea in the west, with the mountains of Midyan, Ash Shifa and Asir forming an escarpment to the east and the provenance for water borne sediments to the wadis along the coastal plain. Since the igneous and metamorphic source rocks are composed of a wide range of minerals including quartz, feldspars, amphiboles, pyroxenes, and micas, it can be assumed that the partially weathered sediments transported to the coastal

plain during flash floods will contain similar minerals, which can in turn be suspended as mineral dust. In contrast the samples collected in Kuwait, Iraq and Afghanistan are from extensive flat lying areas, and to some extent contain minerals such as quartz, calcite, and dolomite from local sedimentary rocks.

Djibouti lies along the African Rift Valley along the west coast of the Gulf of Aden and close to igneous and metamorphic rock formations of the Nubian Plate, separated from the petrographically similar Arabian Plate by the

Red Sea, both regions containing rock formations with substantial amounts of pyroxene, amphibole, and plagioclase.





This at least in part explains the similarity of soils and dust at Djibouti to those along the coastal plain of Saudi Arabia, The mineralogical content of the soils was found to be closely related to the regional geology.

Particle size analysis on the sampled soils showed them to contain too much sand and too little silt to be considered major globally important sources of airborne dust, compared to renowned global sources such as the Bodélé Depression, and silt covered regions of northwest U.S.A. (Engelbrecht et al., 2012;Engelbrecht and Moosmüller, 2014). The low silt content in the investigated samples suggests that the dust plume generated from the Red Sea coastal region is enriched by the coarse dust fraction that deposits quickly. As seen from atmospheric observations, the coastal region is the origin of frequent dust plumes over the Red Sea, probably due to frequent strong wind gusts. These mostly coarse dusts could not be transported the vast distances to the Red Sea and directly deposited there, affecting marine life. Our analysis has revealed that the samples contain compounds of nitrogen, phosphorus and iron that are essential nutrients to marine life (Guerzoni et al., 1997;Migon et al., 2001). The integration of analytical information on dust mineralogy and mineralogical interrelationships, chemistry, and physical properties of soils provides a better understanding of their potential impact on the communities living along the Red Sea (Edgell, 2006;UCAR/NCAR, 2003;Washington et al., 2003). The results from this study can also provide improvements to the input of climate forecasting and dust emission models. The thirteen chemical source profiles will complement those of soil samples collected in other regions of the Middle East (Engelbrecht et al., 2009b), in source attribution studies. Analytical methods developed in this phase of the dust program will be applied for analysis of dust samples deposited from the atmosphere for aerosol characterization studies in the Red Sea coastal region. These will allow further assessing the impact of elevated dust concentrations on regional climate, marine ecology, air quality, and health.

**Data Availability**

The mineralogical and chemical data from this study are available upon request from Georgiy Stenchikov (Georgiy.Stenchikov@kaust.edu.sa).

**Author Contributions**

Georgiy Stenchikov formulated the problem, designed the research project, and supported experimental activities; Johan Engelbrecht advised on aerosol analysis and instrumentation; Weichun Tao defined the dust source areas using satellite observations; Jish Prakash conducted measurements, analysed and combined results; Tahir Yapici and Bashir Warsama helped with instrumentation in the Kaust Core Lab. Prakash, Engelbrecht, and Stenchikov wrote different parts of the paper.

**Acknowledgements**

This research, including the chemical and mineralogical analysis is supported by internal funding from the King Abdullah University of Science and Technology (KAUST). For chemical analyses, this research used the resources





of the KAUST core lab. We acknowledge the contribution from the collaborating laboratories of the RJ Lee Group and Desert Research Institute.

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



**Table1:** Localities of soil sampling sites.

| Site | Proximity | Latitude | Longitude | Elevation (m) |
|------|-----------|----------|-----------|---------------|
| S1 | SE of Al Nasaif | 23.3322° N | 38.9481° E | 94 |
| S2 | SE of Al Nasaif | 23.2961° N | 38.9385° E | 68 |
| S3 | SE of Al Nasaif | 23.2920° N | 38.9100° E | 46 |
| S4 | E of Ar Rayis | 23.5876° N | 38.9243° E | 128 |
| S5 | E of Ar Rayis | 23.5746° N | 38.9213° E | 118 |
| S6 | E of Ar Rayis | 23.5656° N | 38.9193° E | 115 |
| S7 | N of Yanbu | 24.3334° N | 38.0205° E | 113 |
| S8 | N of Yanbu | 24.3239° N | 38.0254° E | 60 |
| S9 | N of Yanbu | 24.3195° N | 38.0245° E | 56 |
| S10 | SW of Mecca | 21.3197° N | 39.5763° E | 128 |
| S11 | SW of Mecca | 21.3232° N | 39.5711° E | 124 |
| S12 | SW of Mecca | 21.3211° N | 39.5593° E | 133 |
| S13 | SW of Mecca | 21.3253° N | 39.5508° E | 118 |





**Table 2:** The volume particle size fraction (%) of the D < 600 μm sieved soil samples.

| Sample | Sand (600–62.5 μm) | Silt (62.5–2 μm) | Clay (< 2 μm) |
|---|---|---|---|
| S1 | 78.0 | 19.2 | 2.8 |
| S2 | 77.2 | 20.5 | 2.3 |
| S3 | 93.3 | 5.7 | 1.0 |
| S4 | 96.3 | 3.0 | 0.7 |
| S5 | 88.4 | 10.0 | 1.7 |
| S6 | 88.5 | 9.8 | 1.6 |
| S7 | 94.3 | 5.2 | 0.5 |
| S8 | 93.5 | 6.0 | 0.5 |
| S9 | 87.1 | 12.1 | 0.9 |
| S10 | 87.8 | 10.6 | 1.6 |
| S11 | 86.6 | 11.4 | 1.9 |
| S12 | 91.1 | 7.6 | 1.2 |
| S13 | 92.7 | 6.1 | 1.2 |





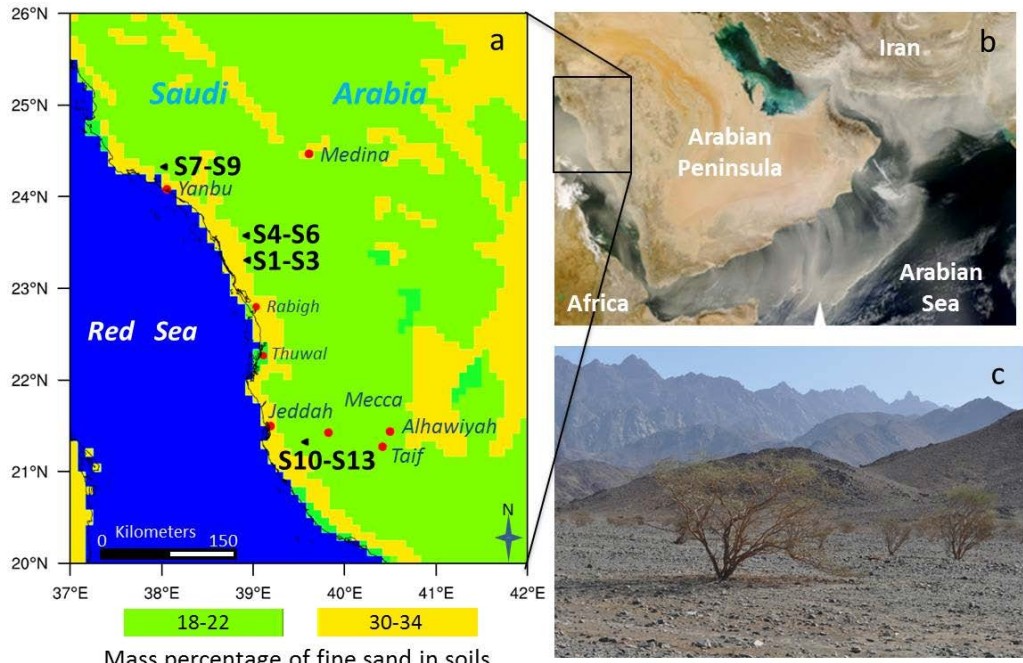

Mass percentage of fine sand in soils

**Figure 1 (a):** Map showing the mass percentage of fine sand in soils, based on STATSGO-FAO soil texture data (Nickovic et al., 2012;Menut et al., 2013), in the Arabian Peninsula, as well as the four localities and thirteen (S1–S13) sampling sites. **(b)** Modis satellite image of dust storm over the Arabian Peninsula captured on February 22, 2008 (NASA Modis web site). **(c)** Sampling site

5    S1 showing the wadi in the foreground with the Hejaz mountain range and escarpment in the distance.





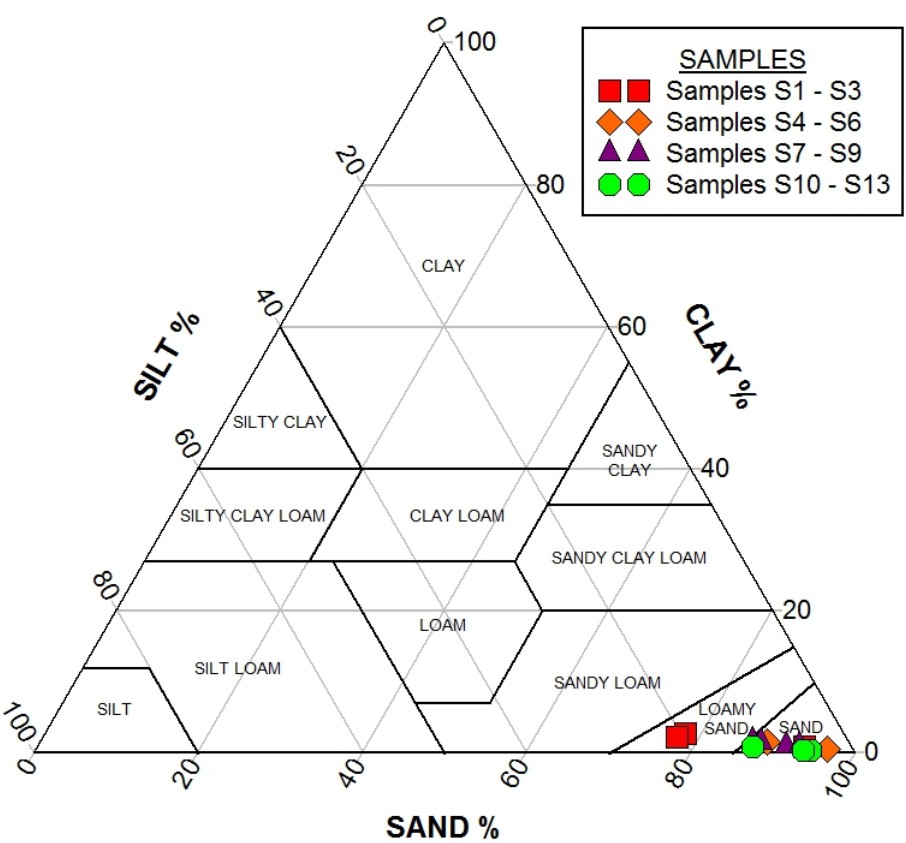

**Figure 2:** US Department of Agriculture (USDA) soil textural triangle showing the grain size plot of the thirteen samples collected for this study. Volume size-class fractions grouped as clay (< 2 µm), silt (2 – 62.5 µm) and sand (62.5 – 600 µm).





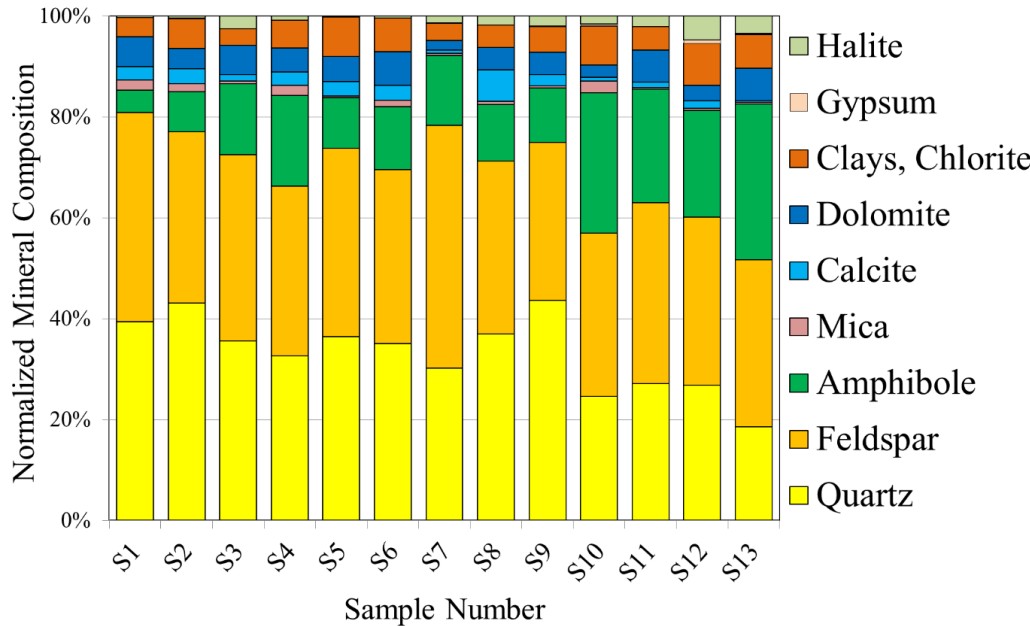

**Figure 3:** Normalized mineral compositions of thirteen D < 38µm sieved soil samples collected at four localities along the Red Sea coastal area, as measured by X-ray diffraction (XRD).



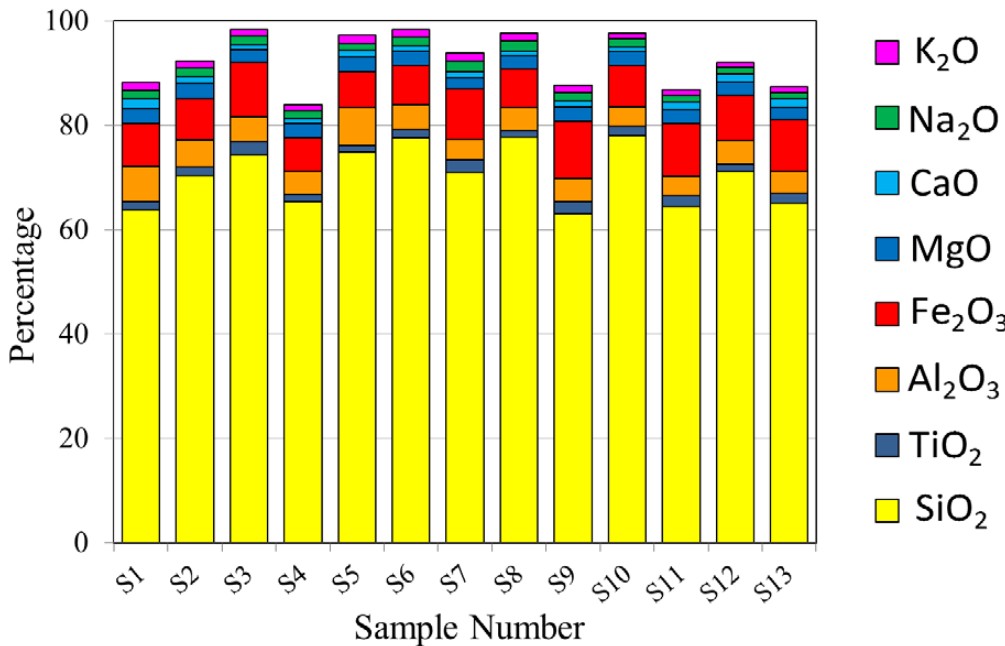

**Figure 4:** Compositional plot showing major oxides from ICP-OES analysis of < 38 µm sieved soils.





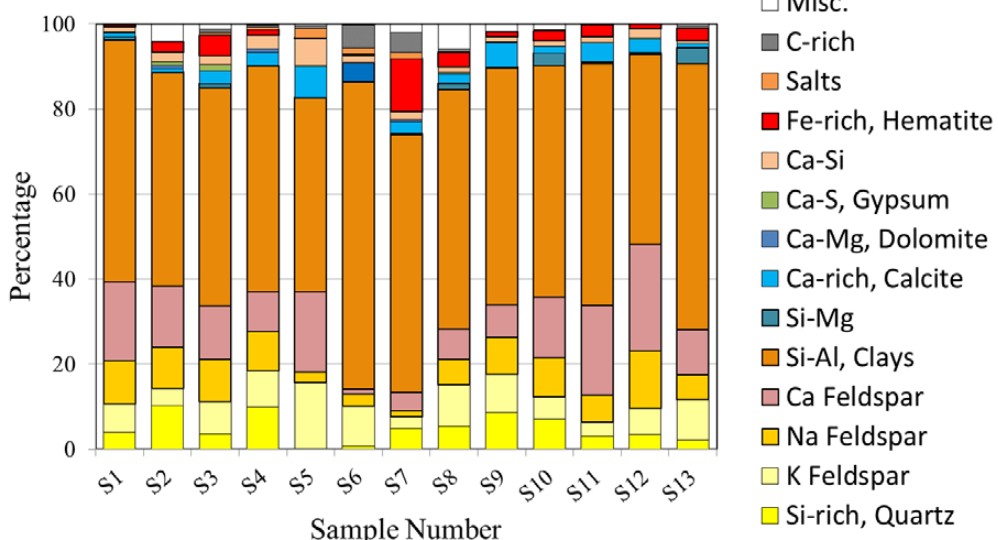

**Figure 5:** CCSEM based individual particle analysis for 0.5 – 38 µm chemical set, with the chemical bins labeled as minerals.





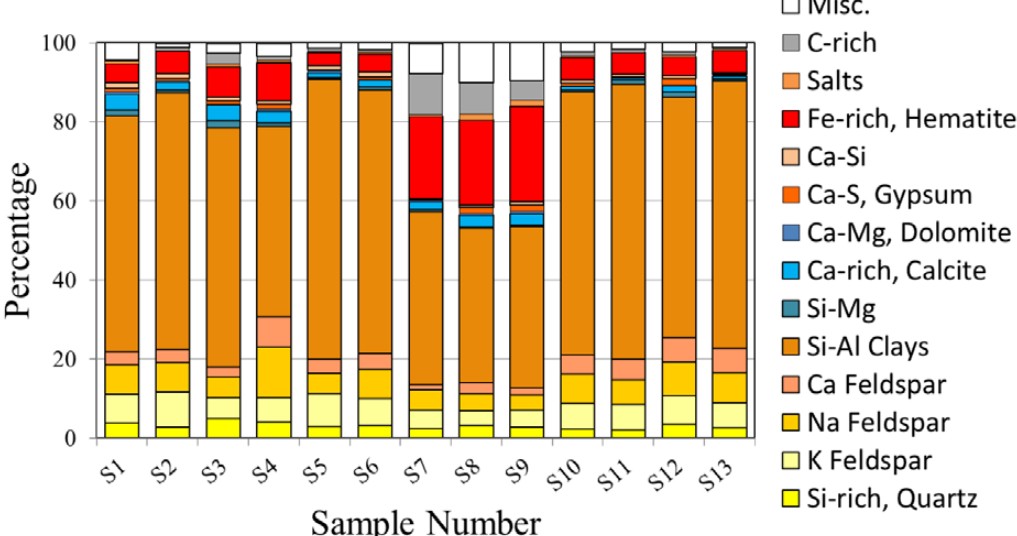

**Figure 6:** CCSEM based individual particle analysis for 0.5 – 2.5 µm (fine) subset, with the chemical bins labeled as minerals.




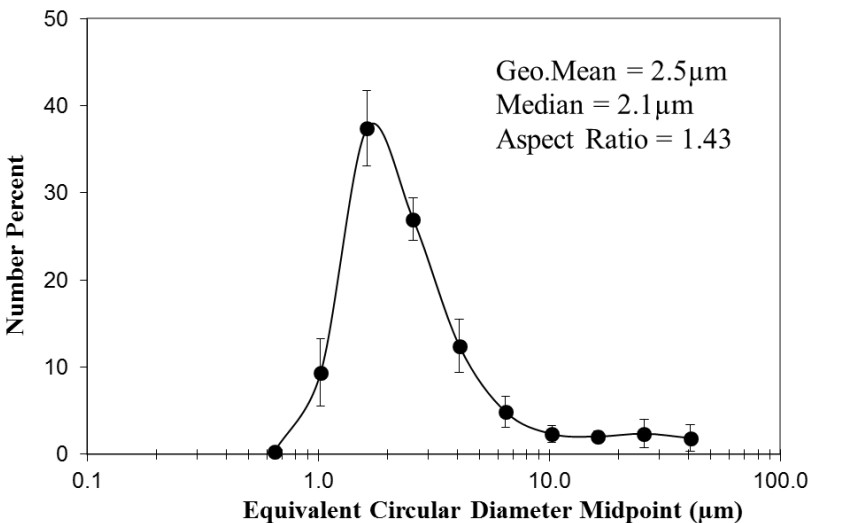

**Figure 7**: Average and standard deviations of particle sizes, as well as size and shape statistics for thirteen D < 38 µm sieved samples, as measured by scanning electron microscopy (CCSEM).




**Appendix B**

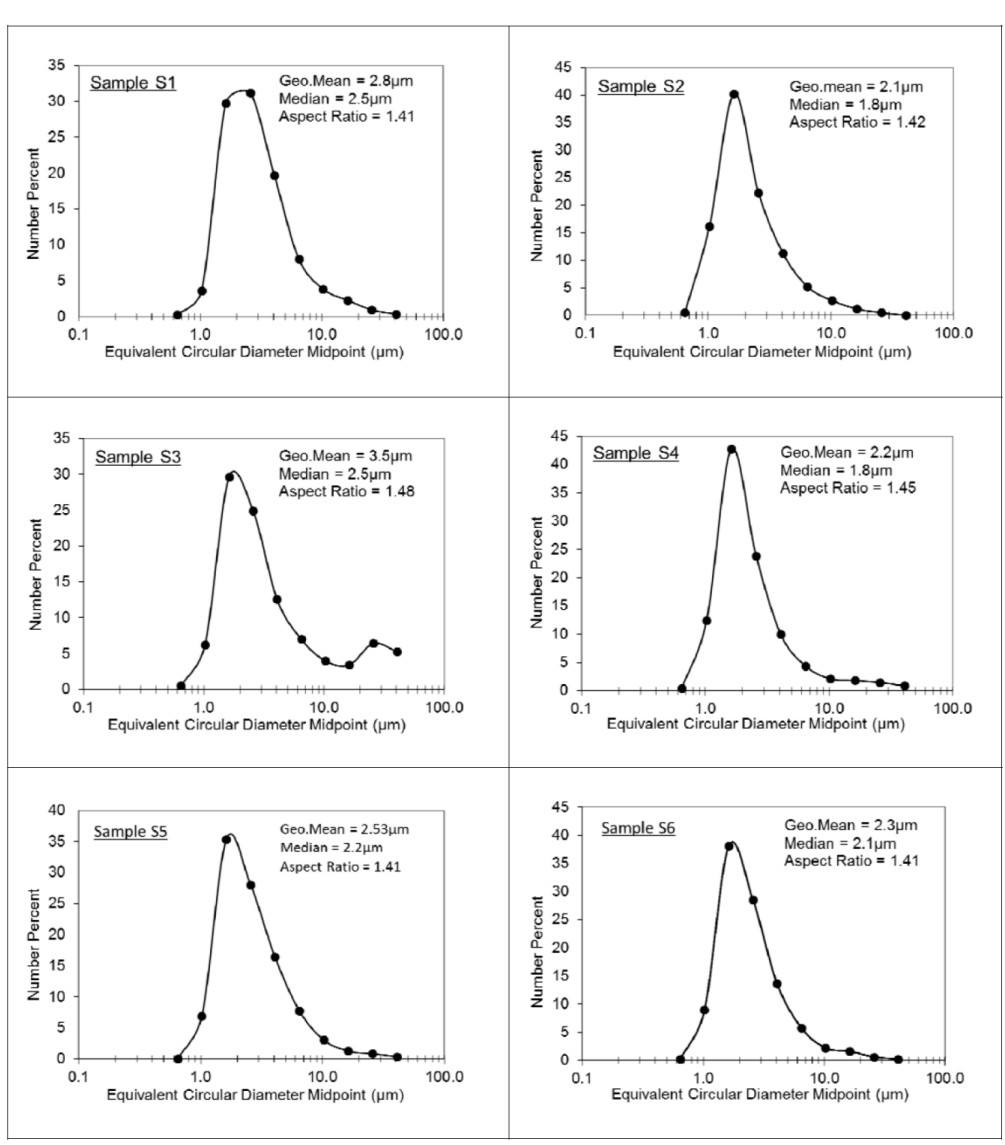

**Figure B1**: Particle size distributions, as well as size and shape statistics for D<38 µm sieved samples S1 – S6, as measured by scanning electron microscopy (SEM).





**Appendix B**

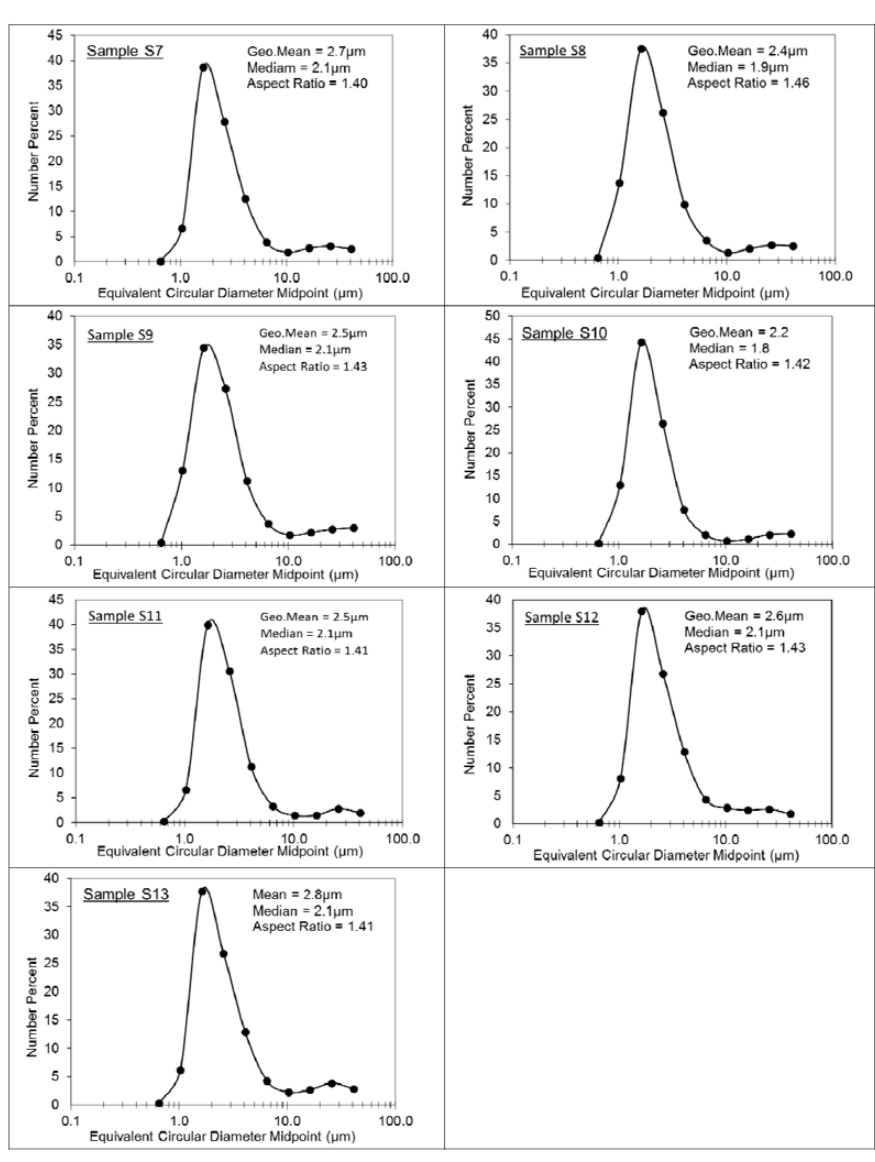

**Figure B2**: Particle size distributions, as well as size and shape statistics for D<38 µm sieved samples S7 – S13, as measured by
scanning electron microscopy (SEM).



**Appendix B**

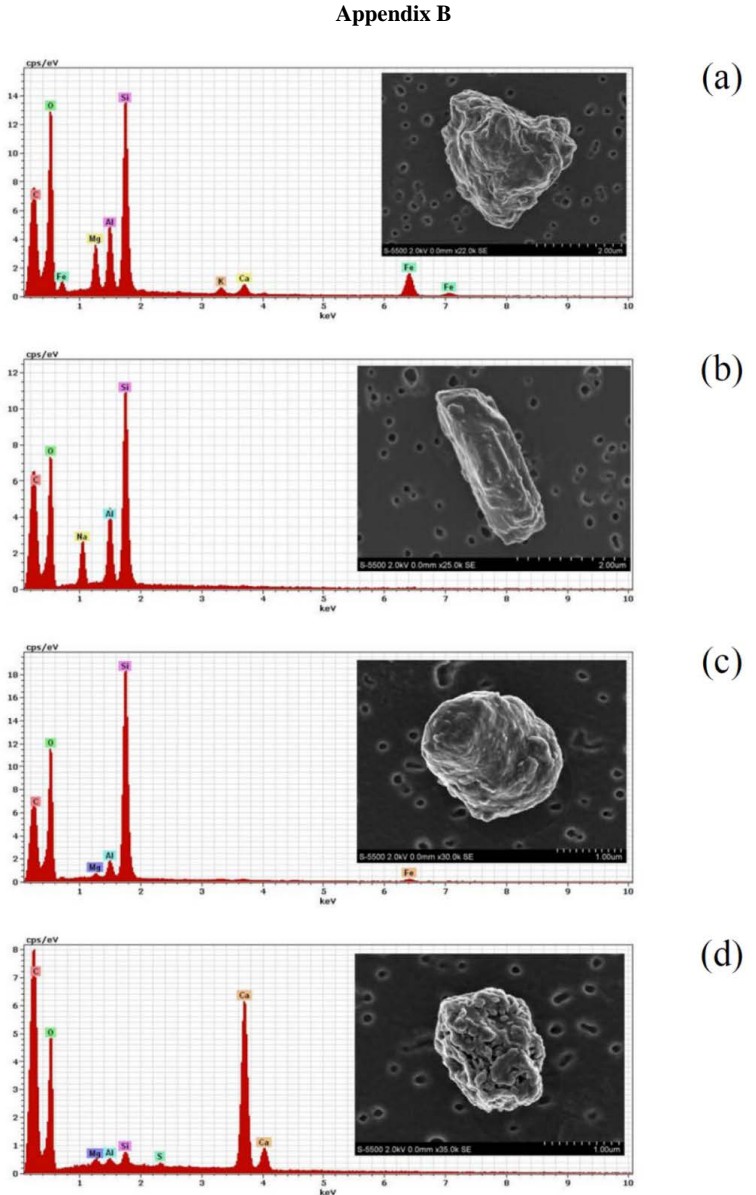

**Figure B3**. Secondary electron images and energy dispersive spectra (EDS) of soil particles **(a)** sample S5, Fe bearing clay mineral possibly illite. **(b)** sample S8, albite feldspar crystal. **(c)** sample S11, rounded quartz grain with minor amount of clay. **(d)** sample S11, cluster of calcite crystals with small amounts of clay and gypsum.





## Appendix C

**Table C1:** Major, minor and trace element compositions by Inductively Coupled Plasma Optical Emission Spectrometry (ICP-OES), and water soluble ions by Ion Chromatography (IC) of grab samples, S1 to S3 collected near Al Nasaif, and S4 to S6 collected near Ar Rayis, all along the Red Sea coastal region. Also tabulated are statistics of the individual particle sizes and morphology as measured by CCSEM.

| Sample # | S1 | | S2 | | S3 | | S4 | | S5 | | S6 | |
|---|---|---|---|---|---|---|---|---|---|---|---|---|
| **Major and minor elements as oxides (%)** | | | | | | | | | | | | |
| | Conc. | Unc. | Conc. | Unc. | Conc. | Unc. | Conc. | Unc. | Conc. | Unc. | Conc. | Unc. |
| $SiO_2$ | 63.795 ± | 3.190 | 70.302 ± | 3.515 | 74.337 ± | 3.717 | 65.436 ± | 3.272 | 74.872 ± | 3.744 | 77.668 ± | 3.883 |
| $TiO_2$ | 1.577 ± | 0.079 | 1.700 ± | 0.085 | 2.536 ± | 0.127 | 1.300 ± | 0.065 | 1.237 ± | 0.062 | 1.511 ± | 0.076 |
| $Al_2O_3$ | 6.768 ± | 0.338 | 5.195 ± | 0.260 | 4.664 ± | 0.233 | 4.367 ± | 0.218 | 7.260 ± | 0.363 | 4.710 ± | 0.236 |
| $Fe_2O_3$ | 8.195 ± | 0.410 | 7.777 ± | 0.389 | 10.497 ± | 0.525 | 6.535 ± | 0.327 | 6.936 ± | 0.347 | 7.584 ± | 0.379 |
| $MnO$ | 0.112 ± | 0.006 | 0.119 ± | 0.006 | 0.135 ± | 0.007 | 0.109 ± | 0.005 | 0.123 ± | 0.006 | 0.144 ± | 0.007 |
| $MgO$ | 2.903 ± | 0.145 | 3.137 ± | 0.157 | 2.478 ± | 0.124 | 2.741 ± | 0.137 | 2.824 ± | 0.141 | 2.767 ± | 0.138 |
| $CaO$ | 1.723 ± | 0.086 | 1.200 ± | 0.060 | 0.895 ± | 0.045 | 0.900 ± | 0.045 | 1.249 ± | 0.062 | 0.909 ± | 0.045 |
| $Na_2O$ | 1.695 ± | 0.085 | 1.577 ± | 0.079 | 1.657 ± | 0.083 | 1.494 ± | 0.075 | 1.248 ± | 0.062 | 1.659 ± | 0.083 |
| $K_2O$ | 1.473 ± | 0.074 | 1.372 ± | 0.069 | 1.269 ± | 0.063 | 1.198 ± | 0.060 | 1.579 ± | 0.079 | 1.484 ± | 0.074 |
| $P_2O_5$ | 0.406 ± | 0.048 | 0.353 ± | 0.047 | 0.400 ± | 0.048 | 0.364 ± | 0.048 | 0.291 ± | 0.046 | 0.403 ± | 0.048 |
| **Total** | 88.649 | | 92.734 | | 98.867 | | 84.444 | | 97.620 | | 98.838 | |
| | | | | | | | | | | | | |
| **Trace elements (ppm)** | | | | | | | | | | | | |
| Li | 17 ± | 1 | 21 ± | 1 | 15 ± | 1 | 19 ± | 1 | 24 ± | 1 | 22 ± | 1 |
| V | 183 ± | 9 | 182 ± | 9 | 242 ± | 12 | 161 ± | 8 | 166 ± | 8 | 191 ± | 10 |
| Cr | 114 ± | 6 | 103 ± | 5 | 150 ± | 8 | 83 ± | 4 | 91 ± | 5 | 98 ± | 5 |
| Co | 30 ± | 1 | 27 ± | 1 | 26 ± | 1 | 29 ± | 1 | 28 ± | 1 | 29 ± | 1 |
| Ni | 55 ± | 3 | 52 ± | 3 | 46 ± | 2 | 45 ± | 2 | 48 ± | 2 | 50 ± | 3 |
| Cu | 29 ± | 1 | 33 ± | 2 | 24 ± | 1 | 42 ± | 2 | 40 ± | 2 | 42 ± | 2 |
| Zn | 39 ± | 2 | 39 ± | 2 | 40 ± | 2 | 39 ± | 2 | 47 ± | 2 | 90 ± | 5 |
| Sr | 294 ± | 16 | 333 ± | 18 | 358 ± | 19 | 288 ± | 15 | 285 ± | 15 | 306 ± | 16 |
| Ba | 318 ± | 16 | 426 ± | 21 | 342 ± | 17 | 408 ± | 20 | 502 ± | 25 | 610 ± | 30 |
| | | | | | | | | | | | | |
| **Water soluble ions (%)** | | | | | | | | | | | | |
| $Mg^{2+}$ | 0.046 ± | 0.001 | 0.038 ± | 0.001 | 0.027 ± | 0.001 | 0.036 ± | 0.001 | 0.044 ± | 0.001 | 0.143 ± | 0.004 |
| $Ca^{2+}$ | 0.171 ± | 0.022 | 0.106 ± | 0.014 | 0.071 ± | 0.009 | 0.107 ± | 0.014 | 0.162 ± | 0.021 | 0.455 ± | 0.058 |
| $Na^+$ | 0.024 ± | 0.001 | 0.005 ± | 0.001 | 0.007 ± | 0.001 | 0.005 ± | 0.001 | 0.008 ± | 0.001 | 0.025 ± | 0.003 |
| $K^+$ | 0.020 ± | 0.002 | 0.010 ± | 0.001 | 0.008 ± | 0.001 | 0.011 ± | 0.001 | 0.025 ± | 0.002 | 0.037 ± | 0.004 |
| $Cl^-$ | 0.092 ± | 0.004 | 0.000 ± | 0.001 | 0.000 ± | 0.001 | 0.000 ± | 0.001 | 0.040 ± | 0.002 | 0.222 ± | 0.010 |
| $SO_4^{2-}$ | 0.078 ± | 0.001 | 0.031 ± | 0.001 | 0.017 ± | 0.001 | 0.126 ± | 0.002 | 0.131 ± | 0.002 | 0.466 ± | 0.007 |
| $PO_4^{3-}$ | 0.002 ± | 0.001 | 0.002 ± | 0.001 | 0.000 ± | 0.001 | 0.000 ± | 0.001 | 0.002 ± | 0.001 | 0.006 ± | 0.004 |
| $NO_3^-$ | 0.018 ± | 0.002 | 0.006 ± | 0.001 | 0.011 ± | 0.001 | 0.016 ± | 0.002 | 0.019 ± | 0.002 | 0.076 ± | 0.007 |
| | | | | | | | | | | | | |
| **Particle diameter from CCSEM measurements (approx. 2000 particles)(μm)** | | | | | | | | | | | | |
| Geom. Mean (μm) | 2.81 | | 2.12 | | 3.50 | | 2.24 | | 2.53 | | 2.35 | |
| Arith. Mean (μm) | 3.66 | | 2.72 | | 6.75 | | 3.25 | | 3.29 | | 3.00 | |
| Skewness | 4.57 | | 4.32 | | 2.34 | | 5.04 | | 5.44 | | 5.51 | |
| Kurtosis | 28.85 | | 25.20 | | 4.63 | | 29.43 | | 40.11 | | 44.00 | |
| | | | | | | | | | | | | |
| Mean aspect ratio | 1.41 | | 1.42 | | 1.48 | | 1.45 | | 1.41 | | 1.41 | |





## Appendix C

**Table C2:** Major, minor and trace element compositions by Inductively Coupled Plasma Optical Emission Spectrometry (ICP-OES), and water soluble ions by Ion Chromatography (IC) of grab samples S7 to S9 collected near Yanbu, and S10 to S13 near Mecca, all along the Red Sea coastal region. Also tabulated are statistics of the individual particle size and morphology as measured by CCSEM.

| Sample # | S7 Conc. | Unc. | S8 Conc. | Unc. | S9 Conc. | Unc. | S10 Conc. | Unc. | S11 Conc. | Unc. | S12 Conc. | Unc. | S13 Conc. | Unc. |
|---|---|---|---|---|---|---|---|---|---|---|---|---|---|---|
| **Major and minor elements as oxides (%)** | | | | | | | | | | | | | | |
| $SiO_2$ | 71.041 ± 3.552 | | 77.76 ± 3.888 | | 62.997 ± 3.150 | | 78.006 ± 3.900 | | 64.44 ± 3.222 | | 71.091 ± 3.555 | | 65.173 ± 3.259 | |
| $TiO_2$ | 2.246 ± 0.112 | | 1.22 ± 0.061 | | 2.401 ± 0.120 | | 1.793 ± 0.090 | | 2.09 ± 0.104 | | 1.499 ± 0.075 | | 1.786 ± 0.089 | |
| $Al_2O_3$ | 4.080 ± 0.204 | | 4.33 ± 0.217 | | 4.351 ± 0.218 | | 3.697 ± 0.185 | | 3.70 ± 0.185 | | 4.516 ± 0.226 | | 4.198 ± 0.210 | |
| $Fe_2O_3$ | 9.563 ± 0.478 | | 7.43 ± 0.371 | | 11.027 ± 0.551 | | 7.997 ± 0.400 | | 10.07 ± 0.504 | | 8.604 ± 0.430 | | 9.936 ± 0.497 | |
| MnO | 0.121 ± 0.006 | | 0.10 ± 0.005 | | 0.156 ± 0.008 | | 0.126 ± 0.006 | | 0.13 ± 0.007 | | 0.115 ± 0.006 | | 0.117 ± 0.006 | |
| MgO | 2.255 ± 0.113 | | 2.53 ± 0.127 | | 2.76 ± 0.138 | | 2.549 ± 0.127 | | 2.62 ± 0.131 | | 2.556 ± 0.128 | | 2.345 ± 0.117 | |
| CaO | 1.109 ± 0.055 | | 1.02 ± 0.051 | | 1.071 ± 0.054 | | 1.064 ± 0.053 | | 1.55 ± 0.077 | | 1.547 ± 0.077 | | 1.586 ± 0.079 | |
| $Na_2O$ | 2.015 ± 0.101 | | 1.92 ± 0.096 | | 1.638 ± 0.082 | | 1.485 ± 0.074 | | 1.31 ± 0.066 | | 1.248 ± 0.062 | | 1.255 ± 0.063 | |
| $K_2O$ | 1.495 ± 0.075 | | 1.49 ± 0.074 | | 1.335 ± 0.067 | | 1.059 ± 0.053 | | 0.96 ± 0.048 | | 0.942 ± 0.047 | | 1.040 ± 0.052 | |
| $P_2O_5$ | 0.467 ± 0.050 | | 0.452 ± 0.049 | | 0.461 ± 0.050 | | 0.385 ± 0.048 | | 0.446 ± 0.049 | | 0.384 ± 0.048 | | 0.384 ± 0.048 | |
| **Total** | 94.392 | | 98.250 | | 88.192 | | 98.160 | | 87.326 | | 92.503 | | 87.819 | |
| **Trace elements (ppm)** | | | | | | | | | | | | | | |
| Li | 16 ± 1 | | 17 ± 1 | | 19 ± 1 | | 14 ± 1 | | 14 ± 1 | | 13 ± 1 | | 12 ± 1 | |
| V | 215 ± 11 | | 157 ± 8 | | 257 ± 13 | | 216 ± 11 | | 283 ± 14 | | 229 ± 11 | | 284 ± 14 | |
| Cr | 129 ± 6 | | 94 ± 5 | | 167 ± 8 | | 142 ± 7 | | 177 ± 9 | | 149 ± 7 | | 171 ± 9 | |
| Co | 26 ± 1 | | 25 ± 1 | | 29 ± 1 | | 31 ± 2 | | 35 ± 2 | | 36 ± 2 | | 32 ± 2 | |
| Ni | 47 ± 2 | | 46 ± 2 | | 53 ± 3 | | 58 ± 3 | | 65 ± 3 | | 61 ± 3 | | 59 ± 3 | |
| Cu | 21 ± 1 | | 22 ± 1 | | 24 ± 1 | | 52 ± 3 | | 55 ± 3 | | 58 ± 3 | | 47 ± 2 | |
| Zn | 41 ± 2 | | 38 ± 2 | | 44 ± 2 | | 41 ± 2 | | 42 ± 2 | | 42 ± 2 | | 39 ± 2 | |
| Sr | 233 ± 13 | | 180 ± 11 | | 381 ± 20 | | 281 ± 15 | | 267 ± 14 | | 259 ± 14 | | 199 ± 11 | |
| Ba | 306 ± 15 | | 302 ± 15 | | 404 ± 20 | | 430 ± 21 | | 409 ± 20 | | 407 ± 20 | | 323 ± 16 | |
| **Water soluble ions (%)** | | | | | | | | | | | | | | |
| $Mg^{2+}$ | 0.024 ± 0.001 | | 0.024 ± 0.001 | | 0.026 ± 0.001 | | 0.025 ± 0.001 | | 0.025 ± 0.001 | | 0.025 ± 0.001 | | 0.028 ± 0.001 | |
| $Ca^{2+}$ | 0.139 ± 0.018 | | 0.138 ± 0.018 | | 0.126 ± 0.016 | | 0.105 ± 0.018 | | 0.061 ± 0.008 | | 0.081 ± 0.010 | | 0.073 ± 0.009 | |
| $Na^+$ | 0.019 ± 0.001 | | 0.012 ± 0.000 | | 0.009 ± 0.001 | | 0.008 ± 0.000 | | 0.009 ± 0.001 | | 0.009 ± 0.001 | | 0.019 ± 0.001 | |
| $K^+$ | 0.016 ± 0.001 | | 0.014 ± 0.001 | | 0.016 ± 0.001 | | 0.016 ± 0.001 | | 0.012 ± 0.001 | | 0.016 ± 0.001 | | 0.018 ± 0.001 | |
| $Cl^-$ | 0.046 ± 0.002 | | 0.037 ± 0.002 | | 0.026 ± 0.001 | | 0.000 ± 0.002 | | 0.000 ± 0.001 | | 0.000 ± 0.001 | | 0.000 ± 0.001 | |
| $SO_4^{2-}$ | 0.088 ± 0.001 | | 0.056 ± 0.001 | | 0.038 ± 0.001 | | 0.091 ± 0.001 | | 0.049 ± 0.001 | | 0.070 ± 0.001 | | 0.063 ± 0.001 | |
| $PO_4^{3-}$ | 0.002 ± 0.001 | | 0.001 ± 0.001 | | 0.000 ± 0.001 | | 0.001 ± 0.001 | | 0.001 ± 0.001 | | 0.001 ± 0.001 | | 0.002 ± 0.001 | |
| $NO_3^-$ | 0.014 ± 0.001 | | 0.009 ± 0.001 | | 0.005 ± 0.001 | | 0.024 ± 0.001 | | 0.012 ± 0.001 | | 0.017 ± 0.002 | | 0.016 ± 0.001 | |
| **Particle diameter from CCSEM measurements (approx. 2000 particles)(μm)** | | | | | | | | | | | | | | |
| Geom. Mean (μm) | 2.68 | | 2.43 | | 2.55 | | 2.21 | | 2.52 | | 2.63 | | 2.82 | |
| Arith. Mean (μm) | 4.50 | | 4.18 | | 4.47 | | 3.67 | | 4.05 | | 4.17 | | 4.94 | |
| Skewness | 3.61 | | 3.83 | | 3.63 | | 4.35 | | 4.21 | | 3.93 | | 3.34 | |
| Kurtosis | 13.38 | | 14.87 | | 13.14 | | 19.06 | | 18.56 | | 16.74 | | 11.20 | |
| Mean aspect ratio | 1.40 | | 1.46 | | 1.43 | | 1.42 | | 1.41 | | 1.43 | | 1.41 | |