# Peer review of "Study of Arabian Red Sea coastal soils as potential mineral dust sources"

_Atmospheric Chemistry and Physics, 2016_

## Referee Comment (RC1) · Anonymous Referee #1 · 25 Apr 2016

The study presents results from measurements of the mineral composition and other properties of soil, based on 13 samples at four locations in the Saudi Arabian coastal plane adjacent to the Red Sea. The region has been understudied so far, although it is an important source of wind blown dust with at least regional impact on human health, climate, and ecosystems. There is a great need for measurements of this kind, not just in the region studied here, but generally, to better understand the impact of dust aerosols as well as to have more data available, which can be used to evaluate and constrain dust aerosols in modeling studies. Thus, I very welcome this study with the new data. The manuscript is generally well written and well structured. Having said this, I see the potential for some improvement in the manuscript, which can be achieved by doing a minor revision. The study should be published after the recommendations have been taken into account.

The authors apply a variety of measurements techniques for studying the mineral properties of the collected soil samples. This is a good approach, since it allows to study the dust mineral properties from different viewpoints. It also reveals, though, that results from the different types of measurements can vary, allowing for ambiguity in the interpretation. This is most evident in the current study where the mineral composition is investigated for the same size range, i.e., $< 38\,\mu$m particle diameter. For instance, the results from the X-ray diffraction (XRD) analysis give a quartz fraction between about 20 and 40 % and a fraction of all the phyllosilicates of not more than 10 %. In contrast, the single particle analysis, using computer controlled scanning electron microscopy, gives a quartz fraction of only up to about 10 %, whereas the phyllosilicates have the largest fraction compared to the other minerals, partially more than 50 %. Which ones of the results from the two different measurement techniques are more reliable? The authors only report these contradicting results next to each other, but a discussion of the significant differences and how to interpret them is lacking. For instance, the possibility of the presence of phyllosilicates in the form amorphous material with poor crystallization is a known source for bias, when XRD analysis is used (*Leinen et al.*, 1994; *Formenti et al.*, 2008; *Kandler et al.*, 2009). Could using this method have caused an overestimation of the quartz fraction? Knowing the answers to such questions would be necessary for properly using the data to constrain or evaluate simulations with dust models.

I recommend following modifications for improving the manuscript:

1. **Section 3, "Sampling and analysis":** For each of the described measurement techniques applied in the study add information about known sources of bias.

2. **Sections 4.3 – 4.5, Figures 3 – 6:** Explicitly state both in the text and in the figures (at the axes or in the captions) the percentages of what variables are shown. Are these the percentages of mass, volume, or number of particles? I

suppose it is the mass fraction in the case of the XRD analysis. It is not clear to me in the cases of the other methods.

3. **Section 5, "Discussion and Conclusions":** Add a discussion of differences in the results from the different measurement techniques and how these differences should be interpreted. How should the data be used, when they are applied in modeling studies?

4. **Section 4.1, Page 7, line 32:** Regarding the statement about the satellite images, I suppose this refers to the two references (Jiang et al. and Kalenderski et al.) that are mentioned elsewhere in the manuscript. Please explicitly reference the two papers once more at the end of the sentence.

**References**

Formenti, P., J. L. Rajot, K. Desboeufs, S. Caquineau, S. Chevaillier, S. Nava, A. Gaudichet, E. Journet, S. Triquet, S. Alfaro, M. Chiari, J. Haywood, H. Coe, , and E. Highwood (2008), Regional variability of the composition of mineral dust from western Africa: Results from the AMMA SOP0/DABEX and DODO field campaigns, *J. Geophys. Res.*, *113*, D00C13, doi:10.1029/2008JD009903.

Kandler, K., L. Schütz, C. Deutscher, M. Ebert, H. Hofmann, S. Jäckel, R. Jaenicke, P. Knippertz, K. Lieke, A. Massling, A. Petzold, A. Schladitz, B. Weinzierl, A. Wiedensohler, S. Zorn, and S. Weinbruch (2009), Size distribution, mass concentration, chemical and mineralogical composition and derived optical parameters of the boundary layer aerosol at Tinfou, Morocco, during SAMUM 2006, *Tellus B*, *61*(1), 32–50, doi:10.1111/j.1600-0889.2008.00385.x.

Leinen, M., J. M. Prospero, E. Arnold, and M. Blank (1994), Mineralogy of aeolian dust reaching the north pacific ocean 1. sampling and analysis, *J. Geophys. Res.*, *99*(D10), 21,017–21,023, doi:10.1029/94JD01735.

---

## Author Comment (AC1) · 31 May 2016

Interactive comment on "Study of Arabian Red Sea coastal soils as potential mineral dust sources" by P. Jish Prakash et al. Anonymous Referee #1 The study presents results from measurements of the mineral composition and other properties of soil, based on 13 samples at four locations in the Saudi Arabian coastal plane adjacent to the Red Sea. The region has been understudied so far, although it is an important source of wind blown dust with at least regional impact on human health, climate, and ecosystems. There is a great need for measurements of this kind, not just in the region studied here, but generally, to better understand the impact of dust aerosols as well as to have more data available, which can be used to evaluate and constrain dust aerosols in modeling studies. Thus, I very welcome this study with the new data. The manuscript is generally well writ-

ten and well structured. Having said this, I see the potential for some improvement in the manuscript, which can be achieved by doing a minor revision. The study should be published after the recommendations have been taken into account. The authors apply a variety of measurements techniques for studying the mineral properties of the collected soil samples. This is a good approach, since it allows to study the dust mineral properties from different viewpoints. It also reveals, though, that results from the different types of measurements can vary, allowing for ambiguity in the interpretation. This is most evident in the current study where the mineral composition is investigated for the same size range, i.e., $< 38\mu$m particle diameter. For instance, the results from the X-ray diffraction (XRD) analysis give a quartz fraction between about 20 and 40 % and a fraction of all the phyllosilicates of not more than 10%. In contrast, the single particle analysis, using computer controlled scanning electron microscopy, gives a quartz fraction of only up to about 10%, whereas the phyllosilicates have the largest fraction compared to the other minerals, partially more than 50 %. Which ones of the results from the two different measurement techniques are more reliable? The authors only report these contradicting results next to each other, but a discussion of the significant differences and how to interpret them is lacking. For instance, the possibility of the presence of phyllosilicates in the form amorphous material with poor crystallization is a known source for bias, when XRD analysis is used (Leinen et al., 1994; Formenti et al., 2008; Kandler et al., 2009). Could using this method have caused an overestimation of the quartz fraction? Knowing the answers to such questions would be necessary for properly using the data to constrain or evaluate simulations with dust models.

I recommend following modifications for improving the manuscript:

1. Section 3, "Sampling and analysis": For each of the described measurement techniques applied in the study add information about known sources of bias.

Authors Comment: Information about known sources of bias will be added to each of the described measurement techniques applied in the study as shown below.

Page 5, Line 23: Minerals with distinctive optical properties, including refractive indices, birefringence, extinction angles, pleochroism, and optical interference patterns, or those showing twinning, distinctive cleavage, and diagnostic extinction angles, can be readily be identified by optical microscopy (Kerr, 1959). Minerals readily identified in these samples by this method include quartz, various feldspars, amphiboles, pyroxenes, micas and carbonates, However, depending on the mineral type, particles <10 $\mu$m in diameter are often difficult to identify by this method, including clay minerals and other layered silicates. The method requires the samples preferably to be mounted in epoxy as a polished thin section. The method is biased towards easily identifiable and coarser minerals, especially those with twinning such as feldspars, and showing color and pleochroism such as hornblende and biotite. The method, although one of the most practical for qualitative mineral analysis, does require mineralogical expertise.

Page 5, Line 31: Powder XRD is particularly suited for fine-grained crystalline mineral mixtures, <10 $\mu$m in diameter. The procedure measures the crystallinity of a sample, i.e. excludes amorphous phases such as glass, partly crystalline layered silicates such as some clays, and hydroxides. If an amorphous phase is present, it will not be fingerprinted by XRD. The assessment of mineral content of a powder sample by the relative intensity ratio (RIR) method suggested by Chung (1974), and as applied in our measurements, does not account for amorphous content.

Page 6, Line 7: This analytical method disperses soil aggregates which are potential dust particles, so shifting the particle size distribution curves towards the smaller particle sizes. This may introduce a bias into the actual size distribution of wind generated dust particles in the field.

Page 6, Line 17: The elemental composition of dust per se does not provide adequate information on its mineral content. However, with a priori knowledge of the mineral composition of the samples, from optical and XRD measurements, "normative" mineral compositions can be calculated. This provides a method for inter-comparing chemically analyzed samples with each other.

Page 7, Line 10: Due to the attenuation of the electron beam as it impinges the particle surface and loss of energy, the analysis is physically limited to an electron interaction volume of 2–5 $\mu$m below the mineral surface, depending on the primary beam voltage and the mineral density (Goldstein et al., 2003). Most of the investigated mineral dust particles have coatings of clay minerals and oxides, which results in an overestimation of the amounts of these minerals when analyzed by CCSEM (Engelbrecht et al., 2009a;Engelbrecht et al., 2016;Engelbrecht et al., 2009b).

2. Sections 4.3 – 4.5, Figures 3 – 6: Explicitly state both in the text and in the figures (at the axes or in the captions) the percentages of what variables are shown. Are these the percentages of mass, volume, or number of particles? I suppose it is the mass fraction in the case of the XRD analysis. It is not clear to me in the cases of the other methods.

Authors Comment: We agree to explicitly state both in the text and in the figures (in the captions) the percentages of variables in section 4.3-4.5 and Figures 3-6.

Section 4.3, Page 8, Line 9: XRD analysis of the thirteen, D<38$\mu$m sieved samples from the Red Sea coastal plain (Fig. 3) confirmed variable mass percentages of quartz (19 − 44%) and feldspars (plagioclase, K-feldspar) (31 − 48%), as well as of amphibole (and pyroxene) (4 − 31%), lesser amounts of calcite (0.4 − 6.2%), dolomite (1.9 − 6.6%), clays and chlorite (smectite, illite, palygorskite, kaolinite) (3.3 − 8.3%), with traces of gypsum (0 − 0.6%) and halite (0.2 − 4.8%).

Section 4.4, Page 8, Line 23 : The sedimentary samples all contain major mass percentages of $SiO_2$, varying between 63% and 78% in the thirteen samples, mostly as the mineral quartz, and lesser mass percentages of $Al_2O_3$ (3.7 − 7.3 %) CaO (0.9 − 1.7 %), $Na_2O$ (1.2 − 2.0 %), and $K_2O$ ( 0.9 − 1.6 %), in plagioclase and potassium feldspars. $SiO_2$ together with $Al_2O_3$, $Fe_2O_3$ (6.5 − 11 %), $TiO_2$ (1.2 − 2.5 %), MnO (0.1− 0.2 %) MgO (2.3 − 3.1 %), and some $K_2O$ (0.9 − 1.6 %) is also contained in the previously identified amphiboles, clays and micas. Small amounts of CaO (0.9 − 1.7%)

are contained in gypsum and calcite, and together with MgO (2.3 – 3.1%), in dolomite.

Section 4.5, Page 9, Line 13: For the total data set, the samples in the 0.5 – 38 $\mu$m size range contain about 0.1 – 10.2% quartz, 5 – 54% feldspar, 45 – 72% clay minerals, as major components with lesser amounts of calcite (0.9 – 7.4 %), dolomite (0 –0.8 %), gypsum (0 –1.5 %), and iron oxides (0.2 –12.4 %). Figure 3: Normalized mineral compositions by percentage of mass [quartz (19 – 44%), feldspars (plagioclase, K-feldspar) (31 – 48%), amphibole and pyroxene (4 – 31%), calcite (0.4 – 6.2%), dolomite (1.9 – 6.6%), clays and chlorite (smectite, illite, palygorskite, kaolinite) (3.3 – 8.3%), gypsum (0 – 0.6%) and halite (0.2 – 4.8%)] of thirteen D < 38$\mu$m sieved soil samples collected at four localities along the Red Sea coastal area, as measured by X-ray diffraction (XRD).

Figure 4: Compositional plot showing major oxides percentages by mass [SiO2 (63 – 78%), TiO2 (1.2 – 2.5 %), Al2O3 (3.7 – 7.3 %), Fe2O3 (6.5 – 11 %), MgO (2.3 – 3.1 %), CaO (0.9 – 1.7 %), Na2O (1.2 – 2.0 %), K2O (0.9 – 1.6 %)] from ICP-OES analysis of < 38 $\mu$m sieved soils.

Figure 5: CCSEM based individual particle analysis for 0.5 – 38 $\mu$m chemical set, with the chemical bins labeled as minerals by mass percentage [Si-rich, Quartz (0.1 – 10.2 %), K Feldspar (2.7 –15.6 %), Ca Feldspar (1.1 – 25 %); Na Feldspar (1.5 – 13.4 %); Si-Al, Clays (44.7 – 72.1 %); Si-Mg (0 – 3.7 %); Ca-Mg, Dolomite (0 – 0.8 %); Ca-Si (0.6 – 6.4 %); Ca-S, Gypsum (0 – 1.5 %); Ca-rich, Calcite (0.9 – 7.4 %); Fe-rich, Hematite ( 0.2 – 12.4 %); Salts (0 – 2.2 %); C-rich (0 – 5.5 %) and Miscellaneous ( 0 – 5.9 %)]

Figure 6: CCSEM based individual particle analysis for 0.5 – 2.5 $\mu$m (fine) subset, with the chemical bins labeled as minerals by mass percentage [Si-rich, Quartz (2.1 – 4.9 %), K Feldspar (3.8 –9.0 %), Na Feldspar (3.8 – 12.9 %); Ca Feldspar (1.4 – 7.7 %); Si-Al, Clays (39.2 – 70.7 %); Si-Mg (0.2 – 1.7 %); Ca-Mg, Dolomite (0 – 0.7 %); Ca-Si (0.3 – 1.5 %); Ca-S, Gypsum (0.1 – 1.7 %); Ca-rich, Calcite (0.6 – 4.1 %); Fe-rich,

Hematite ( 3.2 – 24.1 %); Salts (0.1 – 1.6 %); C-rich (0.4 – 10.5 %) and Miscellaneous ( 1.2 – 10.1 %)].

3. Section 5, "Discussion and Conclusions": Add a discussion of differences in the results from the different measurement techniques and how these differences should be interpreted. How should the data be used, when they are applied in modeling studies?

Authors Comment:

Discussion of differences in the results from the different measurement techniques will be added and interpreted as follows.

Page 9, line 32: The application of a range of techniques for the analysis of properties of soil samples allows for a better understanding of mineral dust. However, the different analytical methods often provide different results, as seen by comparing the XRD, electron microscopy and chemistry of the soils. In this study, the results from the XRD analysis gives a quartz percentage of between about 19 and 44 % and sheet silicates (clays, micas) of between 3 and about 8%. In contrast, the single particle analysis by CCSEM gives a quartz fraction of only up to about 10%, whereas the sheet silicates always have the largest mineral percentage, of up to about 72%. This can lead to ambiguity in the interpretation of the mineralogical composition of the samples. This is evident even where the mineral composition is investigated for the same size range, i.e. $< 38\mu$m particle diameter. Biases in XRD results can be related to the presence of partly amorphous sheet silicates with poor crystallization (Leinen et al., 1994; Formenti et al., 2008; Kandler et al., 2009) and a subsequent overestimation of the quartz fractions. Knowing the answers to such questions would be necessary for properly using the data to constrain or evaluate simulations with dust models. Similarly, the individual particle analysis by CCSEM provides an overestimation of the clay fraction which can be attributed to surface coatings on the quartz and its underestimation (Engelbrecht et al., 2009a, b;Engelbrecht et al., 2016). What is of importance when considering the

application of these results in models, health studies, and remote sensing, is not only the mineralogical composition of the dust, but also their mineralogical interrelationships such as mineral clusters, mineral coatings, and intergrowths.

4. Section 4.1, Page 7, line 32: Regarding the statement about the satellite images, I suppose this refers to the two references (Jiang et al. and Kalenderski et al.) that are mentioned elsewhere in the manuscript. Please explicitly reference the two papers once more at the end of the sentence.

Authors Comment: Two references (Jiang et al., 2009 and Kalenderski et al., 2013) will be added in the text as shown below:

Page 7, Line 32, Section 4.1: However, the satellite images (Jiang et al., 2009 and Kalenderski et al., 2013) show that these coastal dust sources are activated quite frequently.

References:

Chung, F. H.: Quantitative interpretation of X-ray diffraction patterns of mixtures. I. Matrix-flushing method for quantitative multicomponent analysis, Journal of Applied Crystallography, 7, 519-525, doi:10.1107/S0021889874010375, 1974.

Engelbrecht, J. P., McDonald, E. V., Gillies, J. A., Jayanty, R. K. M., Casuccio, G., and Gertler, A. W.: Characterizing mineral dusts and other aerosols from the Middle East – Part 1: Ambient sampling, Inhalation Toxicology, 21, 297-326, 2009a.

Engelbrecht, J. P., McDonald, E. V., Gillies, J. A., Jayanty, R. K. M., Casuccio, G., and Gertler, A. W.: Characterizing mineral dusts and other aerosols from the Middle East – Part 2: Grab samples and re-suspensions, Inhalation Toxicology, 21, 327-336, 2009b.

Engelbrecht, J. P., Moosmüller, H., Pincock, S., Jayanty, R. M., Lersch, T., and Casuccio, G.: Technical Note: Mineralogical, chemical, morphological, and optical interrelationships of mineral dust re-suspensions, Atmospheric Chemistry and Physics, Discussion, 10.5194/acp-2016-286, 2016.

Formenti, P., J. L. Rajot, K. Desboeufs, S. Caquineau, S. Chevaillier, S. Nava, A. Gaudichet, E. Journet, S. Triquet, S. Alfaro, M. Chiari, J. Haywood, H. Coe, and E. Highwood: Regional variability of the composition of mineral dust from western Africa: Results from the AMMA SOP0/DABEX and DODO field campaigns, J. Geophys. Res., 113, D00C13, doi: 10.1029/2008JD009903, 2008.

Goldstein, J., Newbury, D., Joy, D., Lyman, C., Echlin, P., Lifshin, E., Sawyer, L., and Michael, J.: Scanning Electron Microscopy and X-Ray Microanalysis: 3rd Edition, Springer, 689 pp., 2003.

Jiang, H., Farrar, J. T., Beardsley, R. C., Chen, R., and Chen, C.: Zonal surface wind jets across the Red Sea due to mountain gap forcing along both sides of the Red Sea, Geophys. Res. Lett., 36, L19605, 10.1029/2009GL040008, 2009.

Kalenderski, S., Stenchikov, G., and Zhao, C.: Modeling a typical winter-time dust event over the Arabian Peninsula and the Red Sea, Atmos. Chem. Phys., 13, 1999-2014, 10.5194/acp-13-1999-2013, 2013.

Kandler, K., L. Schütz, C. Deutscher, M. Ebert, H. Hofmann, S. Jäckel, R. Jaenicke, P. Knip- pertz, K. Lieke, A. Massling, A. Petzold, A. Schladitz, B. Weinzierl, A. Wiedensohler, S. Zorn, and S. Weinbruch: Size distribution, mass concentration, chemical and mineralog- ical composition and derived optical parameters of the boundary layer aerosol at Tinfou, Morocco, during SAMUM 2006, Tellus B, 61(1), 32–50, doi:10.1111/j.1600-0889.2008.00385.x., 2009.

Kerr, P. F.: Optical Mineralogy, 3rd ed., McGraw-Hill Book Company, Inc., 442 pp., 1959.

Leinen, M., J. M. Prospero, E. Arnold, and M. Blank: Mineralogy of aeolian dust reaching the north Pacific Ocean 1. sampling and analysis, J. Geophys. Res., 99(D10), 21,017–21,023, doi:10.1029/94JD01735, 1994.

[Figure]

Please also note the supplement to this comment:
http://www.atmos-chem-phys-discuss.net/acp-2016-113/acp-2016-113-AC1-supplement.pdf

———————————————————

---

## Referee Comment (RC2) · K. Kandler (Referee) · 28 Jun 2016

The present publication deals with soils in potential dust sources close to the Arabian Sea, an area, of which only few data exists. Mineralogical and geochemical analyses of the potentially windblown size fraction have been investigated. The paper adds new data interesting for atmospheric research.

Similar to the anonymous reviewer, I also would like to see a more critical assessment and comparison of the results from the different techniques. Moreover, a placement of the composition data with regard to other dust source regions would be desirable.

Major remarks

page 4/lines 10-27: The "objectives" chapter apart from first and last sentence, doesn't

really contain any clear objectives, but a mixture of introduction and general information. I suggest rewriting it and clearly stating the goals of the present study. Any introductory information and motivation should go to chapter 1.

9/31: As of now, I suggest terming it rather "Summary and conclusions", as there is not much discussion here.

9/32-10/13: This information belongs rather in introduction. Please merge.

In chapter 4 there is some detailed information of the separate samples, and some intercomparison of the samples. However, I'm somewhat missing a comparison with previous measurements from other regions. Are these sources different to Eastern and Western African, Sahelian, or even Chinese sources? There's for example the reviews of Formenti et al. (2011) and Scheuvens et al. (2013), where data for comparison is readily available, e.g. in terms of mineral and elemental ratios. Or maybe the authors can provide more information on other sources by themselves or use the mentioned databases?

Fig. 3 and 5: Data from SEM and XRD are apparently different, when displayed this way. If SEM data shows particle number percent, I would highly suggest calculating mass percentages from them (by assuming spherical or ellipsoidal particles and assigning an according bulk density) and compare again with XRD data. Differences should be discussed. Is there any chemical fingerprint that can be used to detect amphibole in SEM data?

Minor / corrections

page 2/line 13: It's not just the resolution of the databases limiting statements, but also the general lack of soil data.

2/14-34: The explanation of a source function doesn't seem to contribute to the rest of the manuscript, except for explaining the particle size range of interest. As the latter can be done with a single reference, I suggest removing it.

2/19-29: Please discuss the parameters in order of appearance, and do not jump from one to the other and back.

2/33-34: It seems to me that this motivation sentence should rather be at the beginning of the section.

3/4-26: This is a lot of information, which is difficult to assess for the reader. If you think it is necessary for the present manuscript, I would suggest trying a graphical representation. Otherwise, I suggest restricting it to the information relevant for the current sampling area.

3/27-4/9: I assume this information is from literature. Please add reference(s).

4/16-18: Which observation? Please be more specific and include references, if appropriate.

4-29-5/4: I would assume that precise geographical coordinates would be available for all sampling locations. Please add them, at least to a supplement. That could be done as a table.

5/5-9: This information should be location in the introduction, as it has nothing to do with sampling and analysis.

5/11: Which unwanted artifacts?

5/21-23: General information, omit or place in introduction.

5/25-27: General information, omit or place in introduction.

5/34-6/2: General information, omit or place in introduction.

6/13-15 and 6/19-21: The chemical symbols are sufficient, there is no concern of ambiguity.

6/24-25: General information, omit or place in introduction.

6/33: rastering -> scanning?

7/1: 0.5 $\mu$m < D < 38 $\mu$m

7/29: disaggregation?

8/3: eroded?

9/6: Is it 2000 particles per sample?

9/6-12: On which substrate this analysis was performed, and how were the C-rich identified, if on carbonaceous material?

Fig. B1 and B2: please combine them into a single (or two) color figure(s), as without any grid the small differences are hard to spot. I suggest giving the size and shape statistics as separate table.

Fig. B3: I suggest either removing, as 4 images do not really represent variation in composition and morphology, or making better use of, e.g. by discussion specific details and characteristics of the particles.

Formenti, P., L. Schütz, Y. Balkanski, K. Desboeufs, M. Ebert, K. Kandler, A. Petzold, D. Scheuvens, S. Weinbruch, D. Zhang (2011): Recent progress in understanding physical and chemical properties of mineral dust. Atmos. Chem. Phys. 11, 8231-8256. doi: 10.5194/acp-11-8231-2011

Scheuvens, D., L. Schütz, K. Kandler, M. Ebert, S. Weinbruch (2013): Bulk composition of northern African dust and its source sediments - a compilation. Earth-Sci. Rev. 116, 170-194. doi: 10.1016/j.earscirev.2012.08.005
* * *

---

## Author Response (AR1)

The study presents results from measurements of the mineral composition and other properties of soil, based on 13 samples at four locations in the Saudi Arabian coastal plane adjacent to the Red Sea. The region has been understudied so far, although it is an important source of wind blown dust with at least regional impact on human health, climate, and ecosystems. There is a great need for measurements of this kind, not just in the region studied here, but generally, to better understand the impact of dust aerosols as well as to have more data available, which can be used to evaluate and constrain dust aerosols in modeling studies. Thus, I very welcome this study with the new data. The manuscript is generally well written and well structured. Having said this, I see the potential for some improvement in the manuscript, which can be achieved by doing a minor revision. The study should be published after the recommendations have been taken into account.

The authors apply a variety of measurements techniques for studying the mineral properties of the collected soil samples. This is a good approach, since it allows to study the dust mineral properties from different viewpoints. It also reveals, though, that results from the different types of measurements can vary, allowing for ambiguity in the interpretation. This is most evident in the current study where the mineral composition is investigated for the same size range, i.e., < 38μm particle diameter. For instance, the results from the X-ray diffraction (XRD) analysis give a quartz fraction between about 20 and 40 % and a fraction of all the phyllosilicates of not more than 10%. In contrast, the single particle analysis, using computer controlled scanning electron microscopy, gives a quartz fraction of only up to about 10%, whereas the phyllosilicates have the largest fraction compared to the other minerals, partially more than 50 %. Which ones of the results from the two different measurement techniques are more reliable? The authors only report these contradicting results next to each other, but a discussion of the significant differences and how to interpret them is lacking. For instance, the possibility of the presence of phyllosilicates in the form amorphous material with poor crystallization is a known source for bias, when XRD analysis is used (*Leinen et al.*, 1994; *Formenti et al.*, 2008; *Kandler et al.*, 2009). Could using this method have caused an overestimation of the quartz fraction? Knowing the answers to such questions would be necessary for properly using the data to constrain or evaluate simulations with dust models.

I recommend following modifications for improving the manuscript:

1. **Section 3, "Sampling and analysis":** For each of the described measurement techniques applied in the study add information about known sources of bias.

Authors' Response:

Information about known sources of bias has been added to each of the described measurement techniques applied in the study as shown below.

**Page 5, Line 23:** Minerals with distinctive optical properties, including refractive indices, birefringence, extinction angles, pleochroism, and optical interference patterns, or those showing twinning, distinctive cleavage, and diagnostic extinction angles, can be readily be identified by optical microscopy (Kerr, 1959). Minerals readily identified in these samples by this method include quartz, various feldspars, amphiboles, pyroxenes, micas and carbonates, However, depending on the mineral type, particles <10 μm in diameter are often difficult to identify by this method, including clay minerals and other layered silicates. The method requires the samples preferably to be mounted in epoxy as a polished thin section. The method is biased towards easily identifiable and coarser minerals, especially those with twinning such as feldspars, and showing color and pleochroism such as hornblende and biotite. The method, although one of the most practical for qualitative mineral analysis, does require mineralogical expertise.

**Page 5, Line 31:** Powder XRD is particularly suited for fine-grained crystalline mineral mixtures, <10 μm in diameter. The procedure measures the crystallinity of a sample, i.e. excludes amorphous phases such as clay-like colloids (Formenti et al., 2011;Leinen et al., 1994;Engelbrecht et al., 2016;Kandler et al., 2009), partly crystalline layered silicates such as some clays, and hydroxides. If an amorphous phase is present, it will not be fingerprinted by XRD. The assessment of mineral content of a powder sample by the relative intensity ratio (RIR) method suggested by Chung (1974), and as applied in our measurements, does not account for amorphous content.

**Page 6, Line 7:** This analytical method disperses soil aggregates which are potential dust particles, so shifting the particle size distribution curves towards the smaller particle sizes. This may introduce a bias into the actual size distribution of wind generated dust particles in the field.

**Page 6, Line 17:** The elemental composition of dust *per se* does not provide adequate information on its mineral content. However, with *a priori* knowledge of the mineral composition of the samples, from optical and XRD measurements, "normative" mineral compositions can be calculated. This provides a method for inter-comparing chemically analyzed samples with each other.

**Page 7, Line 10:** Due to the attenuation of the electron beam as it impinges the particle surface and loss of energy, the analysis is physically limited to an electron interaction volume of 2–5 μm below the mineral surface, depending on the primary beam voltage and the mineral density (Goldstein et al., 2003). Most of the investigated mineral dust particles have coatings of clay minerals and oxides, which results in an overestimation of the amounts of these minerals when analyzed by CCSEM (Engelbrecht et al., 2009a;Engelbrecht et al., 2016;Engelbrecht et al., 2009b).

2. **Sections 4.3 – 4.5, Figures 3 – 6:** Explicitly state both in the text and in the figures (at the axes or in the captions) the percentages of what variables are shown. Are these the percentages of mass, volume, or number of particles? I suppose it is the mass fraction in the case of the XRD analysis. It is not clear to me in the cases of the other methods.

Authors' Response:

We agree to explicitly state both in the text and in the figures (in the captions) the percentages of variables in section 4.3-4.5 and Figures 3-6.

**Section 4.3, Page 8, Line 9:** XRD analysis of the thirteen, D<38μm sieved samples from the Red Sea coastal plain (Fig. 3) confirmed variable mass percentages of quartz (19 – 44%) and feldspars (plagioclase, K-feldspar) (31 – 48%), as well as of amphibole (and pyroxene) (4 – 31%), lesser amounts of calcite (0.4 – 6.2%), dolomite (1.9 – 6.6%), clays and chlorite (smectite, illite, palygorskite, kaolinite) (3.3 – 8.3%), with traces of gypsum (0 – 0.6%) and halite (0.2 – 4.8%).

**Section 4.4, Page 8, Line 23 :** The sedimentary samples all contain major mass percentages of $SiO_2$, varying between 63% and 78% in the thirteen samples, mostly as the mineral quartz, and lesser mass percentages of $Al_2O_3$ (3.7 – 7.3 %) CaO (0.9 – 1.7 %), $Na_2O$ (1.2 – 2.0 %), and $K_2O$ ( 0.9 – 1.6 %), in plagioclase and potassium feldspars. $SiO_2$ together with $Al_2O_3$, $Fe_2O_3$ (6.5 – 11 %), $TiO_2$ (1.2 – 2.5 %), MnO (0.1– 0.2 %) MgO (2.3 – 3.1 %), and some $K_2O$ (0.9 – 1.6 %) is also contained in the previously identified amphiboles, clays and micas. Small amounts of CaO (0.9 – 1.7%) are contained in gypsum and calcite, and together with MgO (2.3 – 3.1%), in dolomite.

**Section 4.5, Page 9, Line 13:** For the total data set, the samples in the 0.5 – 38 μm size range contain by mass about 0.1 – 10.2% quartz, 5 – 54% feldspar, 45 – 72% clay minerals, as major components with lesser amounts of calcite (0.9 – 7.4 %), dolomite (0 –0.8 %), gypsum (0 –1.5 %), and iron oxides (0.2 –12.4 %).

**Figure 3**: Normalized mineral compositions by percentage of mass [quartz (19 − 44%), feldspars (plagioclase, K-feldspar) (31 − 48%), amphibole and pyroxene (4 − 31%), calcite (0.4 − 6.2%), dolomite (1.9 − 6.6%), clays and chlorite (smectite, illite, palygorskite, kaolinite) (3.3 − 8.3%), gypsum (0 − 0.6%) and halite (0.2 − 4.8%)] of thirteen D < 38μm sieved soil samples collected at four localities along the Red Sea coastal area, as measured by X-ray diffraction (XRD).

**Figure 4**: Compositional plot showing major oxides percentages by mass [$SiO_2$ (63 − 78%), $TiO_2$ (1.2 − 2.5 %), $Al_2O_3$ (3.7 − 7.3 %), $Fe_2O_3$ (6.5 − 11 %), MgO (2.3 − 3.1 %), CaO (0.9 − 1.7 %), $Na_2O$ (1.2 − 2.0 %), $K_2O$ ( 0.9 − 1.6 %)] from ICP-OES analysis of < 38 μm sieved soils.

**Figure 5**: CCSEM based individual particle analysis for 0.5 − 38 μm chemical set, with the chemical bins labeled as minerals by mass percentage [Si-rich, Quartz (0.1 − 10.2 %), K Feldspar (2.7 −15.6 %), Ca Feldspar (1.1 − 25 %); Na Feldspar (1.5 − 13.4 %); Si-Al, Clays (44.7 − 72.1 %); Si-Mg (0 − 3.7 %); Ca-Mg, Dolomite (0 − 0.8 %); Ca-Si (0.6 − 6.4 %); Ca-S, Gypsum (0 − 1.5 %); Ca-rich, Calcite (0.9 − 7.4 %); Fe-rich, Hematite ( 0.2 − 12.4 %); Salts (0 − 2.2 %); C-rich (0 − 5.5 %) and Misc.( 0 − 5.9 %)]

**Figure 6**: CCSEM based individual particle analysis for 0.5 − 2.5 μm (fine) subset, with the chemical bins labeled as minerals by mass percentage [Si-rich, Quartz (2.1 − 4.9 %), K Feldspar (3.8 −9.0 %), Na Feldspar (3.8 − 12.9 %); Ca Feldspar (1.4 − 7.7 %); Si-Al, Clays (39.2 − 70.7 %); Si-Mg (0.2 − 1.7 %); Ca-Mg, Dolomite (0 − 0.7 %); Ca-Si (0.3 − 1.5 %); Ca-S, Gypsum (0.1 − 1.7 %); Ca-rich, Calcite (0.6 − 4.1 %); Fe-rich, Hematite ( 3.2 − 24.1 %); Salts (0.1 − 1.6 %); C-rich (0.4 − 10.5 %) and Miscellaneous.( 1.2 − 10.1 %)]

3. **Section 5, "Discussion and Conclusions":** Add a discussion of differences in the results from the different measurement techniques and how these differences should be interpreted. How should the data be used, when they are applied in modeling studies?

Authors' Response:

Discussion of differences in the results from the different measurement techniques has been added and interpreted as follows.

**Page 9, line 32:** The application of a range of techniques for the analysis of properties of soil samples allows for a better understanding of mineral dust. However, the different analytical methods often provide different results, as seen by comparing the XRD, electron microscopy and chemistry of the soils. In this study, the results from the XRD analysis gives a quartz percentage of between about 19 and 44 % and sheet silicates (clays, micas) of between 3 and about 8%. In contrast, the single particle analysis by CCSEM gives a quartz fraction of only up to about 10%, whereas the sheet silicates always have the largest mineral percentage, of up to about 72%. This can lead to ambiguity in the interpretation of the mineralogical composition of the samples. This is evident even where the mineral composition is investigated for the same size range, i.e. < 38μm particle diameter. Biases in XRD results can be related to the presence of partly amorphous sheet silicates with poor crystallization (Leinen et al., 1994; Formenti et al., 2008; Kandler et al., 2009) and a subsequent overestimation of the quartz fractions. Knowing the answers to such questions would be necessary for properly using the data to constrain or evaluate simulations with dust models. Similarly, the individual particle analysis by CCSEM provides an overestimation of the clay fraction which can be attributed to surface coatings on the quartz and its underestimation (Engelbrecht et al., 2009a, b;Engelbrecht et al., 2016). What is of importance when considering the application of these results in models, health studies, and remote sensing, is not only the mineralogical composition of the dust, but also their mineralogical interrelationships such as mineral clusters, mineral coatings, and intergrowths.

4. **Section 4.1, Page 7, line 32:** Regarding the statement about the satellite images, I suppose this refers to the two references (Jiang et al. and Kalenderski et al.) that are mentioned elsewhere in the manuscript. Please explicitly reference the two papers once more at the end of the sentence.

Authors' Response:

Two references (Jiang et al., 2009 and Kalenderski et al., 2013) have been added in the text as shown below:

**Page 7, Line 32, Section 4.1:** However, the satellite images (Jiang et al., 2009 and Kalenderski et al., 2013) show that these coastal dust sources are activated quite frequently.

The present publication deals with soils in potential dust sources close to the Arabian Sea, an area, of which only few data exists. Mineralogical and geochemical analyses of the potentially windblown size fraction have been investigated. The paper adds new data interesting for atmospheric research. Similar to the anonymous reviewer, I also would like to see a more critical assessment and comparison of the results from the different techniques. Moreover, a placement of the composition data with regard to other dust source regions would be desirable.

Authors' response:
We added a critical assessment and comparison of the results, as also recommended by Reviewer 1 (see Authors' Response to Reviewer 1). We also added a comparative chemical results from other dust regions

**Major remarks**
page 4/lines 10-27: The "objectives" chapter apart from first and last sentence, doesn't really contain any clear objectives, but a mixture of introduction and general information. I suggest rewriting it and clearly stating the goals of the present study. Any introductory information and motivation should go to chapter 1.

Authors' response:
The introductory information was moved from the objectives to the introductory section. Parts of the objectives section were re-compiled to better describe our objectives

9/31: As of now, I suggest terming it rather "Summary and conclusions", as there is not much discussion here.

Authors' response:
The heading was change to "Summary and conclusions" as suggested.

9/32-10/13: This information belongs rather in introduction. Please merge. In chapter 4 there is some detailed information of the separate samples, and some intercomparison of the samples. However, I'm somewhat missing a comparison with previous measurements from other regions. Are these sources different to Eastern and Western African, Sahelian, or even Chinese sources? There's for example the reviews of Formenti et al. (2011) and Scheuvens et al. (2013), where data for comparison is readily available, e.g. in terms of mineral and elemental ratios. Or maybe the authors can provide more information on other sources by themselves or use the mentioned databases?

Authors' response:
The mentioned text was moved to and re-compiled in the introduction. Comparative chemical results from adjacent regions, together with references were included in section 4.4 as suggested, specifically the Fe/Al ratios. Si/Al, Ca/Al, and Fe/Al ratios are included in the two tables in Appendix A.

Fig. 3 and 5: Data from SEM and XRD are apparently different, when displayed this way. If SEM data shows particle number percent, I would highly suggest calculating mass percentages from them (by assuming spherical or ellipsoidal particles and assigning an according bulk density) and compare again with XRD data. Differences should be discussed. Is there any chemical fingerprint that can be used to detect amphibole in SEM data?

Authors' response:

The SEM results are in mass percentages. The headings of Figures 5 and 6 have been corrected to reflect this correction. We have added a discussed on the differences between the results generated by the different techniques. No chemical finger print for amphibole had been identified from our analyses.

**Minor / corrections**
page 2/line 13: It's not just the resolution of the databases limiting statements, but also the general lack of soil data.

Authors' response:
The sentence was rephrased to include the general lack of soil data

2/14-34: The explanation of a source function doesn't seem to contribute to the rest of the manuscript, except for explaining the particle size range of interest. As the latter can be done with a single reference, I suggest removing it.

Authors' response:
We prefer to retain the source function, since it mentions parameters such as particle size distribution which we analyzed and consider to be important

2/19-29: Please discuss the parameters in order of appearance, and do not jump from one to the other and back.

Authors' response:
We changed the order in which the parameters are discussed so as to be in the order that the appear in the equation

2/33-34: It seems to me that this motivation sentence should rather be at the beginning of the section.

Authors' response:
We moved this to the beginning of the section, as suggested

3/4-26: This is a lot of information, which is difficult to assess for the reader. If you think it is necessary for the present manuscript, I would suggest trying a graphical representation. Otherwise, I suggest restricting it to the information relevant for the current sampling area.

Authors' response:
We deleted the information not specifically close to the sampling region, retaining those on the southwestern part of the Arabian Peninsula.

3/27-4/9: I assume this information is from literature. Please add reference(s).

Authors' response:
We believe the information presented in this part of the paper is sufficiently supported by the existing references

4/16-18: Which observation? Please be more specific and include references, if appropriate.

Authors' response:
This section was moved to the introduction and references added "MODIS and SEVIRI satellite observations"

4-29-5/4: I would assume that precise geographical coordinates would be available for all sampling locations. Please add them, at least to a supplement. That could be done as a table.

Authors' response:
These are listed in Table 1

5/5-9: This information should be location in the introduction, as it has nothing to do with sampling and analysis.

Authors' response:
Moved text to introduction

5/11: Which unwanted artifacts?

Authors' response:
Replaced the word with "detritus"

5/21-23: General information, omit or place in introduction.

Authors' response:
Moved to introduction

5/25-27: General information, omit or place in introduction.

Authors' response:
Moved to introduction

5/34-6/2: General information, omit or place in introduction.

Authors' response:
Moved to introduction

6/13-15 and 6/19-21: The chemical symbols are sufficient, there is no concern of ambiguity.

Authors' response:
For the benefit of non-chemists we prefer to retain the chemical names and symbols, e.g. iron (Fe)

6/24-25: General information, omit or place in introduction.

Authors' response:
Moved to introduction

6/33: rastering -> scanning?

Authors' response:
We retained the word "rastering" as it is used as such in electron microscopy, meaning scanning along a two dimensional grid

7/1: 0.5 μm < D < 38 μm

Authors' response:
Corrected

7/29: disaggregation?

Authors' response:
Corrected to read "disaggregation"

8/3: eroded?

Authors' response:
The spelling was corrected

9/6: Is it 2000 particles per sample?

Authors' response:
We corrected this to read 2000 particles per sample

9/6-12: On which substrate this analysis was performed, and how were the C-rich identified, if on carbonaceous material?

Authors' response:
Polycarbonate substrate was used for the CCSEM analysis. The C rich particles often contain minor amounts of other elements such as sulfur and metals, allowing them to be identified by their backscattered electron image. However, we admit this can be a real challenge and of course impossible to identify if they contain only C.

Fig. B1 and B2: please combine them into a single (or two) color figure(s), as without any grid the small differences are hard to spot. I suggest giving the size and shape statistics as separate table.

Authors' response:
Figure 7 provides an average size distribution and summary statistics. The statistics for individual samples are given in the two tables of Appendix A. Individual sample particle size distribution plots are moved out of the manuscript *per se*, into Supplementary Information section.

Fig. B3: I suggest either removing, as 4 images do not really represent variation in composition and morphology, or making better use of, e.g. by discussion specific details and characteristics of the particles.

Authors' response:
The 4 SEM images are moved out of the manuscript into Supplementary Information section

Formenti, P., L. Schütz, Y. Balkanski, K. Desboeufs, M. Ebert, K. Kandler, A. Petzold, D. Scheuvens, S. Weinbruch, D. Zhang (2011): Recent progress in understanding physical and chemical properties of mineral dust. Atmos. Chem. Phys. 11, 8231-8256. doi: 10.5194/acp-11-8231-2011

Scheuvens, D., L. Schütz, K. Kandler, M. Ebert, S.Weinbruch (2013): Bulk composition
of northern African dust and its source sediments - a compilation. Earth-Sci. Rev. 116,
170-194. doi: 10.1016/j.earscirev.2012.08.005

[revised manuscript text omitted]

---

## Author Response (AR2)

**Arabian Red Sea coastal soils as potential mineral dust sources**

**P. Jish Prakash[1], Georgiy Stenchikov[1], Weichun Tao[1], Tahir Yapici[1], Bashir Warsama[1], and Johann Engelbrecht[2,1]**

[1] King Abdullah University of Science and Technology (KAUST), Physical Science and Engineering Division (PSE), Thuwal, 23955-6900, Saudi Arabia.

[2] Desert Research Institute (DRI), Reno, Nevada 89512-1095, U.S.A.

Correspondence to P. Jish Prakash (jishprakash@gmail.com)

**Co-Editor Decision**

**Co-Editor Decision: Publish subject to technical corrections** (05 Sep 2016) by Paola Formenti
Comments to the Author:
Dear authors, thank you for taking the referees'suggestions into account. The manuscript can be published once you have incorporated the necessary corrections. I urge you to make sure that all the relevant references are included. best regards.

**Authors Response**

We perused the manuscript to ensure that all the references are included. Two corrections were made (see comments below), and we also made other minor corrections throughout the manuscript. The most important are highlighted below.

Regards

[revised manuscript text omitted]